# Cockayne Syndrome Linked to Elevated R-Loops Induced by Stalled RNA Polymerase II during Transcription Elongation

Xuan Zhang[1,2,12], Jun Xu [3,4,5,12], Jing Hu[6,12], Sitao Zhang [7], Yajing Hao[1,8,9,10], Dongyang Zhang[2], Hao Qian[6], Dong Wang [1,2,3] ✉ & Xiang-Dong Fu [11] ✉

Mutations in the Cockayne Syndrome group B (CSB) gene cause cancer in mice, but premature aging and severe neurodevelopmental defects in humans. CSB, a member of the SWI/SNF family of chromatin remodelers, plays diverse roles in regulating gene expression and transcription-coupled nucleotide excision repair (TC-NER); however, these functions do not explain the distinct phenotypic differences observed between CSB-deficient mice and humans. During investigating Cockayne Syndrome-associated genome instability, we uncover an intrinsic mechanism that involves elongating RNA polymerase II (RNAPII) undergoing transient pauses at internal T-runs where CSB is required to propel RNAPII forward. Consequently, CSB deficiency retards RNAPII elongation in these regions, and when coupled with G-rich sequences upstream, exacerbates genome instability by promoting R-loop formation. These R-loop prone motifs are notably abundant in relatively long genes related to neuronal functions in the human genome, but less prevalent in the mouse genome. These findings provide mechanistic insights into differential impacts of CSB deficiency on mice versus humans and suggest that the manifestation of the Cockayne Syndrome phenotype in humans results from the progressive evolution of mammalian genomes.

Cockayne Syndrome (CS) is a severe neurological disorder attributed to mutations occurring in *CSA* or *CSB*[1]. Approximately 70% of CS patients carry truncation or frameshift mutations in the *CSB* gene, which is evolutionarily conserved from yeast to humans[1,2]. Recent studies have shed light on the multifaceted roles of CSB across various cellular systems and pathways, involving chromatin remodeling[3–5], nucleolin regulation[5], rDNA transcription[6,7], redox homeostasis[8,9], and mitochondrial functions[10–13]. As a member of the SWI/SNF family of ATPases, extensive structural and functional studies have revealed a key role of *CSB* in facilitating transcription elongation through nucleosome barriers[14], alongside its involvement in TC-NER[1,15]. CSB binds to elongating or arrested RNAPII to assist the polymerase in

[1]Department of Cellular and Molecular Medicine, University of California San Diego, La Jolla, CA, USA. [2]Department of Chemistry and Biochemistry, University of California San Diego, La Jolla, CA, USA. [3]Department of Pharmaceutical Sciences, Skaggs School of Pharmacy and Pharmaceutical Sciences, University of California San Diego, La Jolla, CA, USA. [4]Genetics and Metabolism Department, The Children's Hospital, School of Medicine, Zhejiang University, National Clinical Research Center for Child Health, Hangzhou, China. [5]The Institute of Translational Medicine, School of Medicine, Zhejiang University, Hangzhou, China. [6]Sichuan Provincial Key Laboratory for Human Disease Gene Study, Sichuan Provincial People's Hospital, University of Electronic Science and Technology of China, Chengdu, China. [7]National Institute of Biological Sciences,7 Science Park Road, Beijing, China. [8]China National Center for Bioinformation, Beijing, China. [9]Beijing Institute of Genomics, Chinese Academy of Sciences, Beijing, China. [10]University of Chinese Academy of Sciences, 100049 Beijing, China. [11]Westlake Laboratory of Life Sciences and Biomedicine, School of Life Sciences and School of Medicine, Westlake University, Hangzhou, Zhejiang, China. [12]These authors contributed equally: Xuan Zhang, Jun Xu, Jing Hu. ✉e-mail: dow003@health.ucsd.edu; fuxiangdong@westlake.edu.cn

overcoming transcriptional barriers or recognizing transcription-blocking lesions to initiate TC-NER[16]. Consequently, *CSB* deficiency elevates the likelihood of RNAPII pausing and/or drop-off during transcription elongation[17,18].

Despite the comprehensive understanding of *CSB* as a critical factor in TC-NER, an elongation factor, and a chromatin remodeler[3–5,15,19–23], the link between CSB deficiency and neurological disorders in humans have remained elusive. The primary involvement of *CSB* in TC-NER fails to explain the disease mechanism, as the disease phenotype emerges without additional DNA damage[24,25]. To date, multiple studies aim to decipher the mechanism of CS at the cellular level. For example, CS has been widely considered as a disease caused by genome instability, leading to a higher chance of cancer development in *CSB*-deficient mice[26]. Additionally, it has been considered a disorder rooted in transcription failure[4,14,22]. However, how such broad functions selectively impact the expression of neuronal-specific genes has remained unclear[27], as mice engineered to carry CS-mimicking *CSB* truncation[26] or null mutation[28] do not develop severe neurological dysfunctions as seen in humans, although these animals did exhibit minor defects such as age-dependent deafness and blindness. Recent efforts in characterizing a rat model carrying a *CSB* truncation mutation unveiled extensive atrophy and demyelination in the cerebellar cortex[29]; however, the full spectrum of CS manifestations is still incompletely mirrored in these animals. Therefore, a pressing problem in the field is to understand how *CSB* deficiency-induced neuropathological phenotype progressively escalated from mice to humans.

In this study, we set out to investigate whether augmented R-loops are the primary cause of genome instability triggered by *CSB* deficiency. Using R-ChIP, a high-resolution R-loop profiling method developed in our lab[30–32], we confirmed that this is indeed the case in *CSB* knockdown (KD) HEK293 cells. To substantiate this discovery, we verified the increase by employing the S9.6 monoclonal antibody, which has a high affinity for RNA-DNA hybrids, and extended our study to multiple human cell types. Notably, we observed a significant increase in induced R-loops within introns, particularly those linked to T-runs positioned downstream of G/C-skewed regions that promote R-loop formation. These specific sequence features are prevalent within the introns of long genes, a characteristic significantly more pronounced in the human genome compared to the mouse genome. Remarkably, the top enriched gene ontology (GO) terms of affected genes in human cells are predominantly related to neural functions, whereas this is not the case in mouse cells. These findings highlight the contribution of non-coding intronic sequences to transcriptional regulation and their important role in the development of human-specific diseases.

## Results

### A dramatically enlarged R-loop landscape in *CSB*-deficient cells
As a key initiator of TC-NER, CSB has been extensively studied for its role in protecting the genome from various DNA-damaging agents, like UV irradiation, reactive oxygen species, double-strand DNA break (DSB)[33,34]. R-loops, resulting from the invasion of the nascent RNA into the DNA bubble, are typically associated with dynamic pausing and pause/release of RNAPII at promoter regions under physiological conditions[35]. However, R-loops are dramatically induced to cause genome instability under various pathological conditions[36]. Considering the key role of *CSB* in maintaining genome stability, we hypothesized that *CSB* might play a central role in regulating the formation and resolution of R-loops. To test this hypothesis, we utilized engineered HEK293 cells expressing a V5-tagged catalytically inactive RNaseH1, as previously developed in our lab[30,32], and conducted duplicated R-ChIP experiments in mock-treated and *CSB* KD cells without introducing additional insults to the genome.

Efficient *CSB* KD with two independent siRNAs did not show any visible impact on cell morphology or growth, while robust immunoprecipitation of RNaseH1 was achieved using the anti-V5 antibody (Supplementary Fig. 1a, b). Through our enhanced data analysis procedure (see Methods, Supplementary Fig. 1c, d), we were able to capture the full spectrum of R-loops with a typical median length of ~180 nt. As expected, the identified R-loops displayed characteristic GC-skewed sequences upstream of the ensemble R-loop summit (Supplementary Fig. 1e). Given the high reproducibility observed in duplicated experiments under each experimental condition (Supplementary Fig. 1f, g), we merged the data from the replicates for downstream analysis.

Upon comparing peak intensity before and after *CSB* KD, it immediately became evident that the intensity of most R-loops increased in response to *CSB* KD (Fig. 1a). Specific examples displaying strand-specific signals on both plus (+, up) and minus (−, bottom) strands before (gray peaks) and after *CSB* KD (yellow- or blue-colored peaks) further illustrate this augmentation (Fig. 1b and Supplementary Fig. 1g). Quantitatively, *CSB* KD led to a remarkable up to threefold increase in R-loop peak number (from 2775 to 7750, Fig. 1c), accompanied by a slight rise in peak size (from a median of ~180 nt to ~210 nt, Fig. 1d). This escalation of R-loops due to *CSB* deficiency was also reflected by the increased global signal intensity (Fig. 1e and Supplementary Fig. 1h) and confirmed by staining with S9.6, a monoclonal antibody specifically targeting RNA-DNA hybrids (Supplementary Fig. 2a, b). Interestingly, elevated R-loops were detected not only near transcription start sites (TSSs), where RNAPII frequently undergoes pausing and pause/release, but also in genebodies (GBs), transcription termination sites (TTSs), and other intergenic regions (Fig. 1f). These findings thus reveal the broad participation of *CSB* in regulating transcription elongation across the genome, either by modulating RNAPII pausing and pause/release at TSSs, joining the existing regulatory mechanisms, and/or inducing de novo R-loop formation within specific genic and intergenic regions in the human genome.

### Frequent CSB KD-induced R-loops associated with T-runs within genebodies
To understand the molecular basis for inducing R-loops in various genomic regions, we conducted motif analysis separately on common (2261 peaks detected in both siNC and siCSB-treated cells) and gained (5489 peaks newly induced by siCSB) R-loops (Fig. 1g, middle). R-loops are well-known to associate with G/C-skewed sequences on the non-template strand, which is thought to promote the creation of DNA bubbles for nascent RNAs to invade due to the tendency of these sequences to form G-quadruplex structures[37]. This feature is clearly associated with both common and gained R-loops (Supplementary Fig. 3a). As G/C-skewed sequences typically reside upstream of the R-loop summit, we further characterized *CSB* KD-induced R-loops by segregating sequences into "head" (upstream of the R-loop summit) and 'tail' (downstream). Interestingly, while the head sequences exhibit a prevalent enrichment of G/C-skewed motifs in both common and gained R-loops (Fig. 1g), the tail sequences, particularly those associated with gained R-loops, display an additional feature of T-runs (Fig. 1g, right). We observed that the most enriched motif in the tail of the gained R-loop is still G/C-skewed, which should be the part of an extended G/C-skewed sequence from the head and a T-run is always located further downstream (Supplementary Fig. 3b).

To further characterize the motif distribution, we focused on the two major groups of R-loops, one at TSSs and the other within GBs, conducting separate motif analysis. This revealed that both common and gained R-loops at TSSs are largely characterized by G/C-skewed sequences (Supplementary Fig. 3d). While this also holds true for common R-loops detected within GBs, we found that T-enriched motifs are uniquely associated with those gained within GBs in response to *CSB* KD (Supplementary Fig. 3d, right). These observations suggest that *CSB* may join the existing mechanisms to regulate

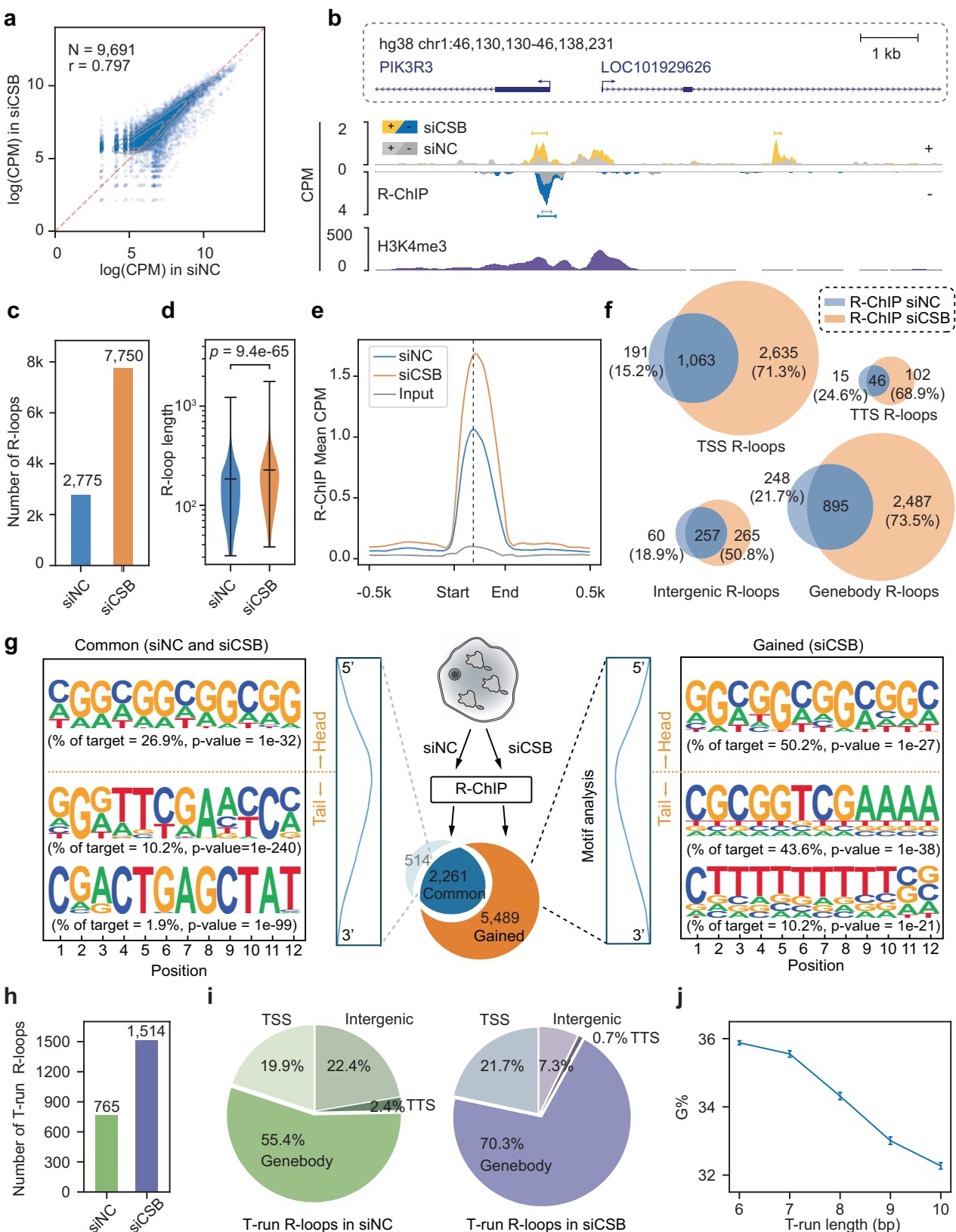

RNAPII pausing/release at TSSs and certain GB regions. Moreover, once the polymerase transitions into productive elongation, this chromatin remodeler becomes particularly important for over-coming transcription blockage at T-runs within GBs, which have been functionally linked to RNAPII backtrack and/or drop-off during elongation[38,39].

## Characterizing genebody R-loops featuring G-rich sequences and T-runs

Focusing on the gained R-loops within GBs, we observed that G/C-skewed motifs are mostly represented by G-rich sequences, which are well-known to have a high propensity to form G-quadruplex[40]. To further delineate these motif characteristics, each mapped R-loop was

**Fig. 1 | Depletion of CSB enhances R-loop formation. a** Scatter plot of R-ChIP, showing R-loop signals in mock-treated (siNC; X-axis) versus CSB-depleted HEK293 cells (siCSB; Y-axis). Each dot represents an R-loop peak region. The scale for both X-axis and Y-axis is log(CPM + 1); CPM: count per million. **b** Genome browser tracks showing a region containing R-loop signals from R-ChIP and public H3K4me3 signals on both the up (+) strand (gray and yellow) and lower (−) strand (gray and blue) in siNC-treated (gray) and siCSB-treated (yellow or blue) HEK293 cells. Gene annotations are shown on top of the tracks. **c** R-loop number profile under siNC and siCSB treatment conditions. **d** R-loop length distribution under siNC and siCSB treatment conditions. Statistical significance was assessed using a two-tailed Mann–Whitney *U*-test. **e** Metaplot of R-loop signals in siNC (blue) and siCSB (orange) treated HEK293 cells. Signals are centered on the R-loop summit on ±0.5 kb surrounding regions. **f** Venn diagrams of R-loops at TSSs (transcription start sites), Genebodies, TTSs transcription termination sites, and Intergenic regions. **g** Motifs enriched on R-loops. Motif enrichment in the head half and tail half was separately analyzed. Along with each enriched motif is the frequency and associated *p*-value. Shared and siCSB-induced R-loops are displayed on the left and right, respectively. **h, i** The number of T-run associated R-loops detected in siNC and siCSB-treated HEK293 cells (**h**) and the distribution of their genomic locations (**i**). **j** G percentage in relationship with the length of T-run on individual R-loops. The plot shows the average G percentage in non-template DNA vs the length of T-run in common T-run-associated R-loops. Error bars indicate the standard error of the mean (*n* = 172, 61, 59, 52, and 69 common T-run-associated R-loops, from left to right).

assigned a G/C-skewed score, a G percentage (G%), and a T percentage (T%) for comparison between common (blue), gained (orange) R-loops, and a similar set of randomly selected sequences (gray) from protein-coding genes. Interestingly, both the GC-skew score and G% show stronger associations with R-loops compared to random sequences within GBs, whereas the opposite is true with T-runs (Supplementary Fig. 3c). Consistent with the enriched T-run motif within induced R-loops in GBs, we found that 765 T-run-associated R-loop peaks were already detectable before *CSB* KD, and the number increased to approximately 1500 after *CSB* KD (Fig. 1h). The majority of these T-run-associated common R-loops are situated in GBs (55.4%), a percentage that becomes more pronounced (70.3%) in response to *CSB* KD (Fig. 1i). Quantitatively, when segregating the length of T-runs into different bins, G% exhibited a negative correlation with the length of continuous T-runs (Fig. 1j). Altogether, these observations suggest that high G% sequences coupled with T-runs in specific GB regions are particularly susceptible to R-loop formation in a *CSB*-dependent manner. Furthermore, while T-runs alone do not trigger R-loop formation, they do help reduce the reliance on G%.

The requirement for an upstream G-rich sequence to couple with a downstream T-run to trigger R-loop formation explains why functional 3′ splice sites (3′ss), characterized by a polypyrimidine tract, typically do not induce R-loops. To illustrate this, we calculated the distance of each T-run-associated R-loop to the nearest 3′ss and found that most of those mapped R-loops are >0.5 kb away from functional 3′ss (Supplementary Fig. 4a). This indicates that 3′ss are not hotspots for R-loop formation, likely because of the lack of upstream G-rich sequences (Supplementary Fig. 4b). Additionally, the involvement of nascent RNA containing functional 3′ss in co-transcriptional splicing may also hinder their annealing to DNA. These may underlie the fact that little R-loop signals are detected at 3′ss, even in *CSB* KD cells (Supplementary Fig. 4c). Together, these observations suggest that CSB potentially regulates RNAPII pausing/release not only at TSSs but also within GBs to minimize R-loop formation, especially in critical genic regions where a G-rich sequence couples with a downstream T-run, creating significant barriers to transcription elongation.

### Colocalization of CSB and RNAPII at R-loops during transcription elongation

To substantiate the direct contribution of *CSB* to RNAPII pausing and pause-induced R-loop formation during productive elongation within GBs, we next compared R-loop formation in these regions with the physical binding events of CSB. To this end, we conducted genome-wide ChIP-seq to identify CSB's action sites. Due to insufficient IP efficiency with existing anti-CSB antibodies, we established a stable HEK293 cell line expressing Flag-tagged *CSB* (Supplementary Fig. 5a). This cell line exhibited approximately a fivefold increase in *CSB* expression, detectable with the anti-Flag antibody (Supplementary Fig. 5b, c). Analysis of the highly reproducible ChIP-seq data (Supplementary Fig. 5d) revealed CSB binding occurrences at TSSs (31.6%), GBs (44.9%), TTSs (9.6%), and intergenic regions (14.0%) (Supplementary Fig. 5e). Upon alignment of R-loops within GBs before and after *CSB* KD (Fig. 2a, column 1, 2), it became apparent that CSB binding coincides with mapped R-loops (Fig. 2a, column 3), as further illustrated using a representative gene example (Fig. 2b). This suggests the direct participation of CSB in modulating R-loop formation during transcription elongation.

To assess the impact of R-loop formation and CSB activity on elongating RNAPII, we conducted RNAPII ChIP-seq before and after *CSB* KD. Using the highly reproducible libraries constructed under both conditions (Supplementary Fig. 6a–c), we focused on GBs for our metagene analysis and found that RNAPII was transiently paused at CSB binding sites before *CSB* KD (Fig. 2a, column 4). Upon *CSB* KD, we observed a significant decrease in RNAPII ChIP-seq signals, particularly towards the 3′ end (Fig. 2a, column 5). This indicates that instead of leading to further accumulation of paused RNAPII, the induction of R-loops caused a significant degree of RNAPII drop-off, which is in line with the data on the analysis of nascent RNA in *CSB* KD cells[17]. To provide further evidence for pausing-induced RNAPII drop-off, we conducted PRO-seq[41] assays to measure transcriptionally engaged RNAPII before and after *CSB* KD (Supplementary Fig. 6d). As anticipated, PRO-seq signals were significantly reduced in response to *CSB* KD (Fig. 2a, column 6, 7). Collectively, these data provide evidence for RNAPII pausing followed by a degree of drop-off in R-loop forming regions within GBs, which may constitute a key mechanism for the regulation of gene expression (see below).

We noted that CSB and RNAPII ChIP-seq peaks are similar, which appear slightly larger than the size of R-loops in the heatmap (compare columns 1 and 2 with columns 3 to 5) (Fig. 2a). This indicates that CSB may act together with RNAPII during transcription elongation, which agrees their direct interaction from structural analyses[19]. Considering that T-runs are known to cause RNAPII backtracking[38], we further aligned measured ChIP-seq and PRO-seq signals at T-runs and observed a tendency for RNAPII to pause at two positions: one upstream of the R-loop and the other at the summit of the R-loop (Fig. 2c). Interestingly, we noted a subsequent decline in PRO-seq signals shortly after RNAPII pausing at both locations (Fig. 2c), exemplified by a specific gene instance (Fig. 2d). These observations reinforce the possibility for a degree of pausing-triggered RNAPII drop-off, thereby freeing the 3′ end of nascent RNA to drive R-loop formation.

### Direct evidence for CSB-enhanced RNAPII elongation through T-runs

To provide direct evidence for T-run induced RNAPII pausing and the involvement of *CSB* in this process, we took a reductionist approach by using purified RNAPII complex from yeast to conduct in vitro transcription run-off assays on DNA templates with or without a G-rich sequence coupled with a T-run. For this purpose, we selected two genic regions based on our in vivo RNAPII ChIP-seq and R-ChIP data. One region served as a control without detectable RNAPII pausing (not shown); the other contained two distinct G-rich clusters (blue box), each followed by a T-run (red) (Fig. 3a). With this in vitro transcription system, we first assembled the purified RNAPII complex on a DNA template strand (TS) annealed with a radioactively labeled RNA primer.

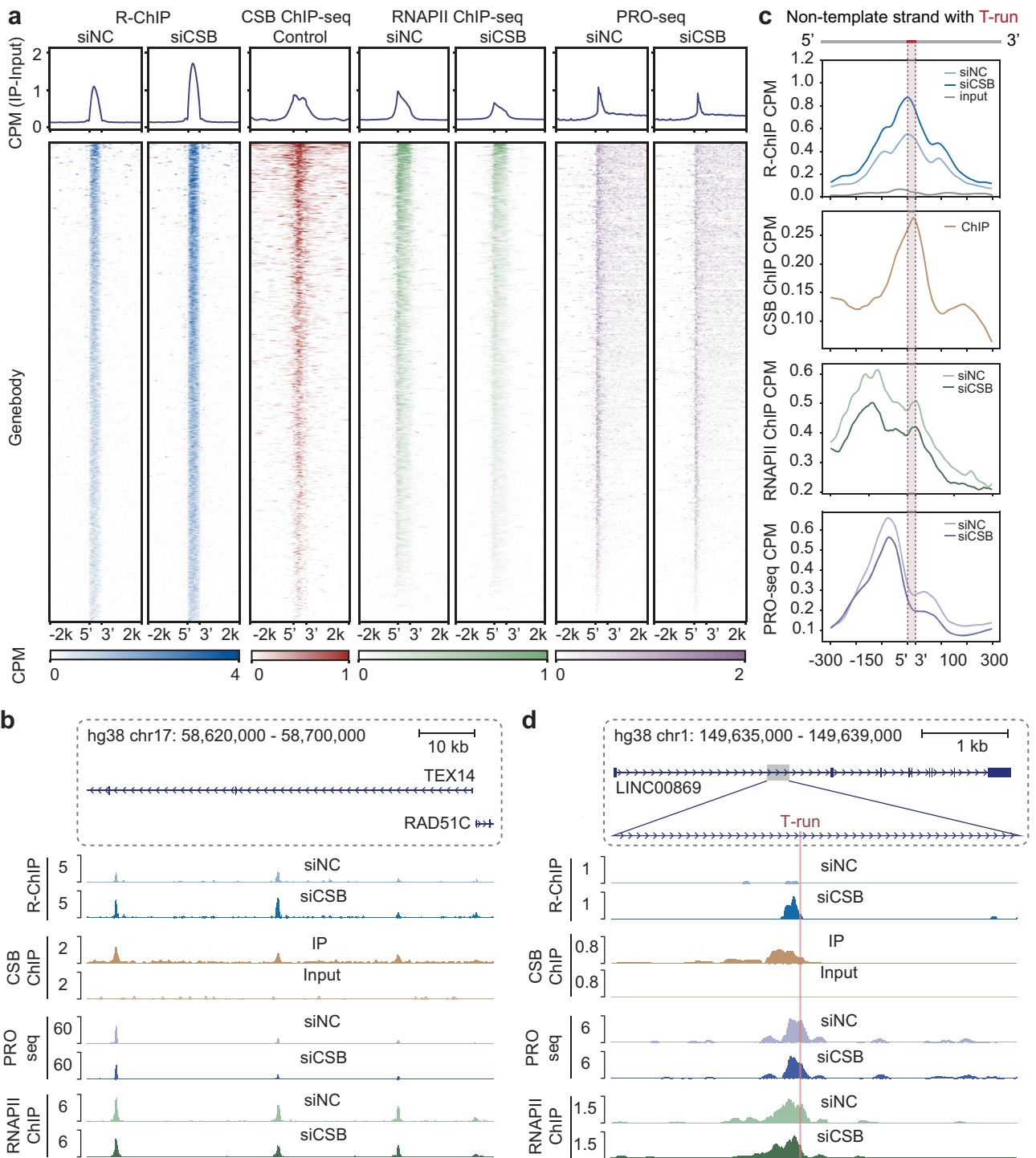

**Fig. 2 | Depletion of CSB promotes R-loop formation at T-run. a** Heatmap showing the pattern of R-loops, RNAPII binding, and PRO-seq signals before and after CSB depletion in genebody regions. CSB binding is also profiled under control conditions. The color scales reflect signal intensity. **b** Genome browser tracks showing the signals at R-loops located in the *TEX14* genebody under siNC and siCSB conditions. R-ChIP signals (top) detected in siNC and siCSB are compared with CSB binding (Middle 1), PRO-seq signal (Middle 2), and RNAPII ChIP-seq signals (Bottom). **c** Meta profile at T-run R-loops. Meta R-ChIP-seq signals under siNC and siCSB conditions are shown by blue lines; Meta CSB ChIP-seq signals by brown lines; Meta RNAPII ChIP-seq signals by green lines; Meta PRO-seq signals by purple lines. Signals are centered on the T-run at each R-loop with 0.3 kb surrounding regions. **d** Genome browser tracks show the signals at a T-run associated R-loop located in the LINC00869 genebody under siNC and siCSB conditions. Shown are induced R-loop (top) in comparison with CSB binding in siNC-treated cells (Middle 1), reduced PRO-seq signals (Middle 2), and reduced RNAPII bindig (Bottom).

Subsequently, we paired it with a biotinylated non-template strand (NTS) to form a transcription bubble within the polymerase (Fig. 3b). The pre-assembled RNAPII complex, ready for elongation, was then immobilized on beads and ligated to double-stranded DNA templates with or without G-rich sequences linked to T-runs. After washing, the primer extension was initiated from the radiolabeled primer by the addition of NTPs.

Both DNA templates produced the expected full-length RNA run-off product of approximately 220 nt. With the control construct, we detected two minor RNAPII pausing events: one occurring between

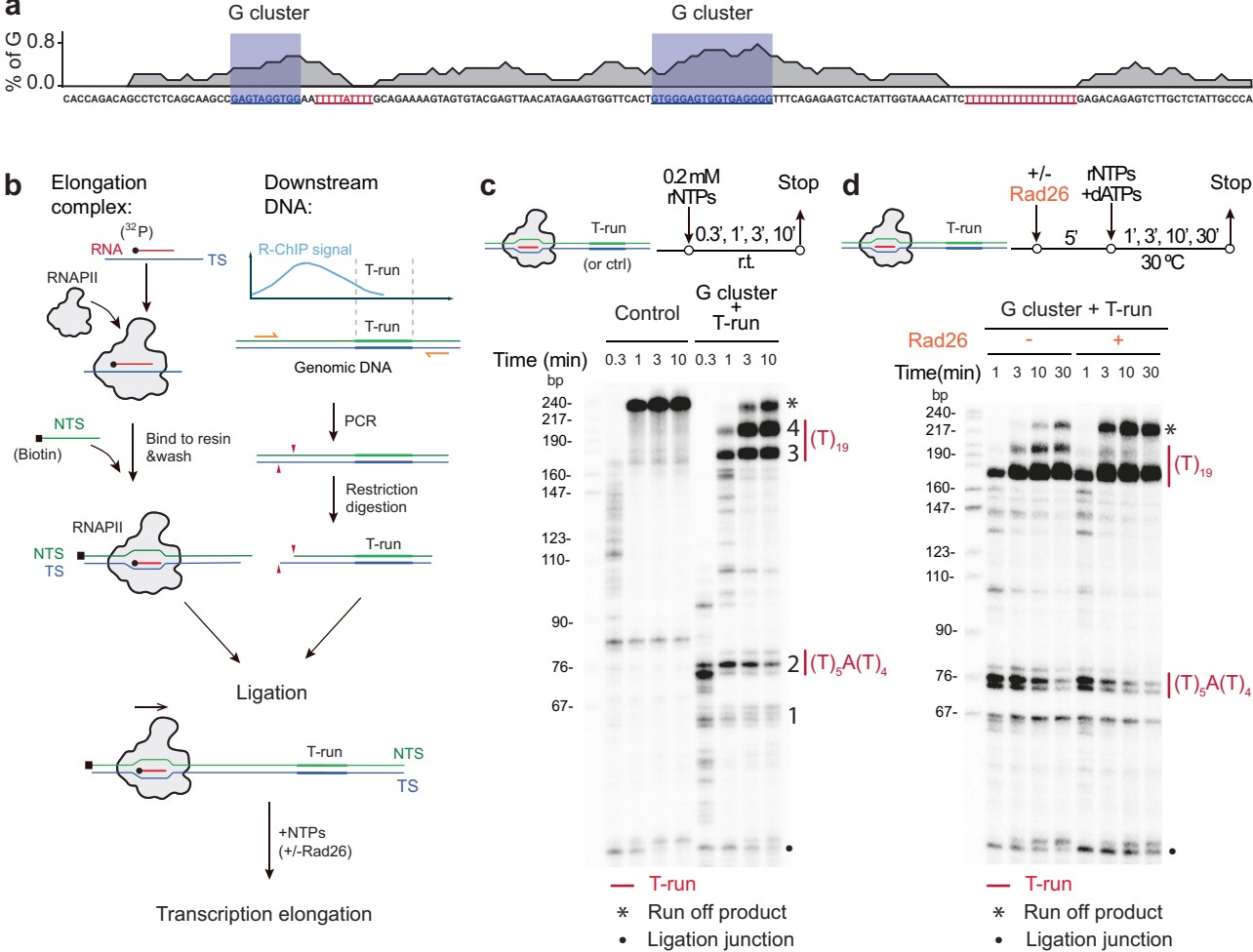

**Fig. 3 | CSB resolves RNAPII Pausing at T-runs. a** Sequence feature of the DNA template used for the run-off assay. Two G clusters are shaded in blue box and Two T-runs are highlighted in red. **b** A scheme of the experimental setup to study the function of CSB in RNAPII transcription in vitro. An RNAPII elongation complex (EC) segment and a T-run containing DNA fragment (based on the R-loop identified from R-ChIP) were reconstituted individually and ligated by T4 ligase. A biotin label is shown as a black square. **c** RNAPII pausing at T-run. The top section displays a schematic of the transcription reaction. Below shows the run-off signals, illustrating multiple pausing sites during RNAPII transcription on a control sequence lacking

T-runs (Left) compared to a template with T-run associated R-loops (Right). The pausing sites and T-run regions are indicated on the right. Ligation truncation corresponds to transcripts from reconstituted ECs that failed to ligate with the downstream T-run associated R-loop template. **d** Role of Rad26 in promoting RNAPII bypassing the T-run. The top section depicts a schematic of the transcription reaction. Below shows specific RNAPII pausing sites during RNAPII elongation along a T-run associated R-loop template without (left) or with (right) purified Rad26. Experiments in **c**, **d** were repeated independently three times with similar results.

76 bp to 90 bp and the other located near the common ligation junction region in the bottom (Fig. 3c, left). Significantly, we detected two minor (Band 1 and 2) and two major (Band 3 and 4) clusters of RNAPII pausing events associated with $T_5AT_4$ and $T_{19}$-runs (Fig. 3c, right). Upon the addition of purified Rad26 (yeast ortholog of CSB) to the in vitro transcription reaction, we detected a significant reduction of RNAPII pausing signals, particularly within the middle of the T19-run (Band 4), accompanied by an increase in the full-length run-off product (Fig. 3d). In contrast, we observed that Band 3 was relatively less sensitive to Rad26 (Fig. 3d). Quantification based on three independent experiments reinforced this observation (Supplementary Fig. 7a, b). These results suggest that while it may be more challenging for Rad26 to push forward paused RNAPII if it is arrested in the front of a robust R-loop; once the polymerase has progressed halfway, the additional force provided by Rad26 becomes more effective in enabling RNAPII to fully overcome the barrier. Overall, despite variable efficiency in different regions, these experiments demonstrated a direct role of CSB in facilitating paused RNAPII to overcome the adverse effects of specific R-loop prone sequence motifs during transcription elongation.

## Elongation blockage by T-run induced R-loop formation and transcription backtrack

The established in vitro transcription system enabled us to directly investigate the functional interplay between RNAPII pausing and R-loop formation. We first asked whether a specific RNAPII pausing event could be functionally linked to R-loop formation, the latter of which would be sensitive to RNase H treatment. For this purpose, we conducted an in vitro transcription reaction for 30 min followed by the addition of RNaseH1 to digest R-loop associated RNA at different time intervals (0, 1, 2, and 5 min) (Fig. 4a). Under this experimental scheme, the control template showed negligible effect upon RNaseH1 treatment, indicating the absence of non-specific nuclease activities with the recombinant RNaseH1. However, the template containing the G-cluster/T-run was sensitive to RNaseH1 treatment in a time-dependent manner at both $T_5AT_4$ and $T_{19}$-runs (Fig. 4a, red lines). In addition to the observed decrease in RNA band intensity at the R-loop region, we also captured various shorter fragments as digestion products (Fig. 4a, blue lines). Notably, the intensity of these shorter fragments increased with longer RNaseH1 treatment,

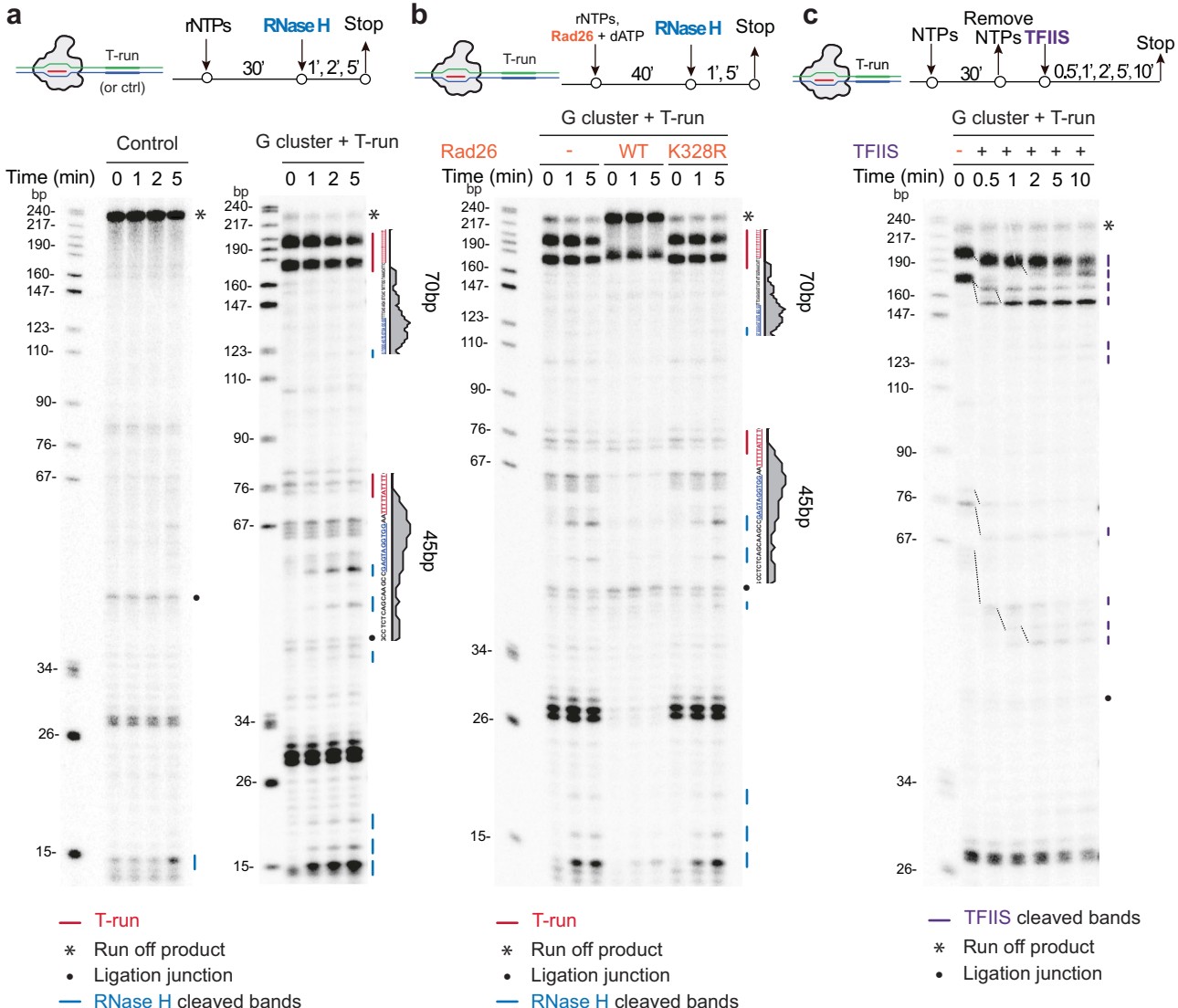

**Fig. 4 | CSB prevents R-loop formation induced by paused RNAPII at T-run.**
**a** R-loop induced by T-run during transcription. Top: a schematic of the transcription reaction. Below: RNaseH1-mediated cleavage of high molecular weight transcripts from the T-run containing template, but not control template. Blue lines indicate RNase H-digested transcripts, while the black dot denotes RNA transcripts resulting from a template with incomplete ligation (ligation truncation). **b** Prevention of R-loop formation by Rad26. Top: a schematic of the transcription reaction. Below: RNaseH1-mediated cleavage of transcripts arrested at T-runs during RNAPII transcription in the absence of Rad26 (left), in the presence of the WT Rad26 (Middle), or an ATPase-deficient mutant rad26 (Right). **c** RNAPII backtracking during in vitro elongation. Top: a schematic of the transcription reaction. Below: the TFIIS assay conducted on the T-run containing template during various time points of RNAPII transcription. Experiments in **a**–**c** were repeated independently three times with similar results.

specifically at the location corresponding to the upstream border of the R-loop where a major peak of RNAPII ChIP-seq signals was detected (see Fig. 2d). These findings provided direct evidence that a G-cluster coupled with a T-run was indeed facilitated R-loop formation, subsequently impeding RNAPII elongation across the R-loop.

As described in the previous section, Rad26 was highly effective in pushing paused RNAPII forward in the middle of the major $T_{19}$-run (Band 4), with a relatively minor impact on RNAPII stalled at the front of such $T_{19}$-run (Band 3) (Fig. 3c, d). To further explore the impact of Rad26 on R-loop formation under these conditions, we conducted additional tests and found that the presence of Rad26 potently eliminated the bands sensitive to RNaseH1 (note that the bands due to RNAPII pausing remained unaffected) (Fig. 4b). Furthermore, we demonstrated that the ATP-dependent translocase activity of Rad26 is essential for preventing R-loop formation. These findings suggest that RNAPII is largely tied up by the R-loops formed at these pausing sites,

and Rad26 utilizes its ATP-dependent translocase activity to propel RNAPII forward through those barriers.

It is well known that when RNAPII is transiently paused, it tends to undergo backtracking, leading to cleavage of the nascent RNA stimulated by TFIIS[38]. To determine whether R-loop-induced pausing of RNAPII around T-runs might undergo backtracked in the absence of CSB, we introduced TFIIS into our in vitro transcription run-off reactions. We found that paused RNAPII near the T-runs could all quantitatively backtrack along the DNA template by approximately 10 to 30 bp (Fig. 4c). Under this experimental setting, we detected little RNAPII drop-off since TFIIS treatment does not affect RNA signals resulting from RNAPII drop-off events. Together, these in vitro transcription elongation assays established a series of causal events, including (i) G-cluster/T-run-induced formation of R-loops, (ii) differential impact on RNAPII pausing in a position-sensitive manner, (iii) pausing-induced RNAPII backtrack in the presence of TFIIS, but not RNAPII drop-off, the latter of which could only be efficiently detected

in *CSB*-deficient cells, as indicated by decreased RNAPII ChIP-seq signals in response to *CSB* KD[42] (see also Fig. 2a, c).

## Potential molecular basis for CS-associated neuronal defects in humans

Having established the molecular basis for R-loop-induced RNAPII pausing in *CSB*-deficient cells and the role of CSB in R-loop formation, we then explored how these molecular discoveries could be leveraged to explain the phenotypic defects linked to a defective general chromatin remodeler. For this purpose, we noted that our cellular model–HEK293 cells engineered to express V5-tagged RNaseH1–is suitable for studying the impact on the expression of many neuronal-specific genes, as this embryonic kidney-derived cell line expresses numerous neuronal-specific genes and HEK293 cells have their capacity to form synapses with co-cultured neurons[43]. We, therefore examined the gene expression profile of HEK293 cells before and after *CSB* KD. Given the primary impact of *CSB* deficiency on nascent RNA production, we conducted cell fractionation, focusing on nascent RNAs that remained associated with chromatin, which contained twice the amount of intronic reads compared to exonic reads (Fig. 5a and Supplementary Fig. 8a–c). Our analysis identified both upregulated and downregulated genes following *CSB* KD (Fig. 5b). For instance, the *RAD51B* gene showcased altered expression, while the nearby *ACTN1* and *ZFYVE26* genes remained unaffected (Supplementary Fig. 8d).

Importantly, we found that downregulated genes, but not upregulated ones, are significantly associated with *CSB* KD-induced R-loops, indicating the direct impact of R-loop formation and associated RNAPII pausing/drop-off on the expression of those downregulated genes (Fig. 5c). Strikingly, gene ontology (GO term) analysis revealed that while the upregulated genes are related to general cellular activities, which might result from the compensatory responses (Supplementary Fig. 8e), those downregulated ones are primarily linked to neuronal functions and DNA damage responses (Fig. 5d). Interestingly, we further noted that most of these downregulated genes are long genes that tend to express and function in the central nervous system[44] (Fig. 5e, see those highlighted in Fig. 5b), even though the general population of long genes represents only a small fraction of the human genome (Supplementary Fig. 8f). These observations suggest a mechanistic link between *CSB* deficiency, R-loop formation, and the expression of relatively long genes.

Because the *CSB* KD-induced RNAPII elongation defects are directly linked to R-loop formation within genic regions exhibiting specific G%, T%, and GC-skew characteristics, we took advantage of this observation to distinguish R-loop prone regions from random sequences (Fig. 5f, left). We developed a binary classifier using the support vector machine (SVM) learning algorithm, which achieved 86% accuracy in identifying R-loop-prone regions based on our R-ChIP data (Fig. 5f, right). As expected, the predicated T-run linked R-loops showed a significant association with downregulated genes, but not upregulated ones (Fig. 5g). We next applied this predictive tool to examine two sets of public RNA-seq data: one from human bone osteosarcoma epithelial U2OS cells before and after *CSB* KD (Supplementary Fig. 9a, b)[45] and another from CS patient-derived fibroblasts (CS1AN), which were either rescued by wild-type *CSB* cDNA or left untreated (Supplementary Fig. 9c, d)[7]. We found that the downregulated genes in response to *CSB* KD in HEK293 cells were consistently represented in both of these cellular systems (Supplementary Fig. 9a, c). Importantly, these downregulated genes were characterized by longer gene lengths and a higher propensity for R-loop formation at T-run regions (Supplementary Fig. 9b, d).

We further investigated the effects of *CSB* KD on R-loop levels and gene expression in a neuronal cell model where human dermal fibroblasts (HDFs) were trans-differentiated into functional neurons by depleting the RNA-binding protein PTBP1, as detailed in our previous work[46]. These HDFs closely resemble CS1AN cells, as both originate

from human skin fibroblasts. Our findings from these differentiated HDFs closely align with our observations in HEK293 cells, including elevated R-loop signals detected with S9.6 and the selective downregulation of relatively long genes (Supplementary Fig. 10a–e). To further corroborate these findings, we took advantage of a recently deposited RNA-seq dataset on cerebral organoids derived from one healthy and two CS patient-derived induced pluripotent stem cells (iPSCs)[47]. We observed that the downregulated genes in both *CSB*-deficient neurons compared to health control were much longer in comparison with the upregulated ones (Supplementary Fig. 10f).

## Comparative analysis of *CSB* deficiency effects on human and mouse cells

We next wished to understand the molecular basis for the differential impact of CSB deficiency in human versus mouse cells[26,28]. Analyzing the RNA-seq data from the kidney of *CSB* null mice[28], we found that gene length did not strongly associate with either up- or downregulated genes (Fig. 5h, i). Additionally, enriched GO terms were unrelated to neural functions among either up- or downregulated genes (Supplementary Fig. 11). This may be because most mouse analogs of long genes in the human genome are considerably shorter (Fig. 5j). Consequently, the predicated T-run-associated R-loops are not prevalently linked to either up- or downregulated genes (Fig. 5k).

We further compared the impact of *CSB* KD on two neuronal cell lines, i.e. N2A (mouse) and SH-SY5Y (human) cells. To monitor *CSB* KD before and after the induction of neuronal differentiation on these cells, we transduced lentiviral vectors expressing Tet-inducible control shGFP or shCSB into these cells (Fig. 6a), and upon the addition of Dox, we achieved efficient *CSB* KD in these cell lines (Supplementary Fig. 12a, b). We next asked whether *CSB* KD impacted retinoic acid (RA)-induced differentiation and observed little difference in the development of morphology and viability with or without *CSB* KD on these cell lines under our experimental conditions (Supplementary Fig. 12c–e). These observations are in line with the relatively normal development of *CSB* null mice[26] and efficient neuronal differentiation of CS patient-derived ES cells[48].

To substantiate our key findings with respect to R-loop formation and its differential impact on the expression of relatively long genes related to neural functions, we conducted a direct comparison between RA-differentiated N2A and SH-SY5Y cells. Dox efficiently induced *CSB* KD in these differentiated cells (Supplementary Fig. 12a, b). Interestingly, Dox-induced *CSB* KD significantly prompted R-loop formation in differentiated SH-SY5Y cells, but notably absent in N2A cells, as evidenced by S9.6 staining and quantitative analysis (Fig. 6b–e). RNA-seq analysis revealed that the impact of *CSB* KD on human SH-SY5Y cells was much stronger than on mouse N2A cells, as indicated by a ~8-fold difference in $\log_{10}P$ values and >10-fold difference in the number of affected genes (Fig. 6f, g). *CSB* KD selectively downregulated relatively long genes in SH-SY5Y cells (Fig. 6h), but not in N2A cells (Fig. 6i). Concurrently, only those downregulated genes in SH-SY5Y cells were enriched with GO terms linked to various neural pathways (Fig. 6j, k). These observations suggest that divergent genome evolution as the underlying molecular basis for the phenotypic difference induced by *CSB* deficiency in mice versus humans.

## Discussion

### Sensing a T-run by elongating RNAPII during transcription elongation

Initial NET-seq experiments suggested transient RNAPII pausing at functional 3'ss[49], but subsequent studies argued that such signals might result from splicing intermediates that also contain a 3'-OH group for ligation to a 5'-phosphorylated linker, which was eliminated in the presence of a splicing inhibitor[50,51]. More recently, it was further argued that RNAPII was paused in the front of functional 3'ss because

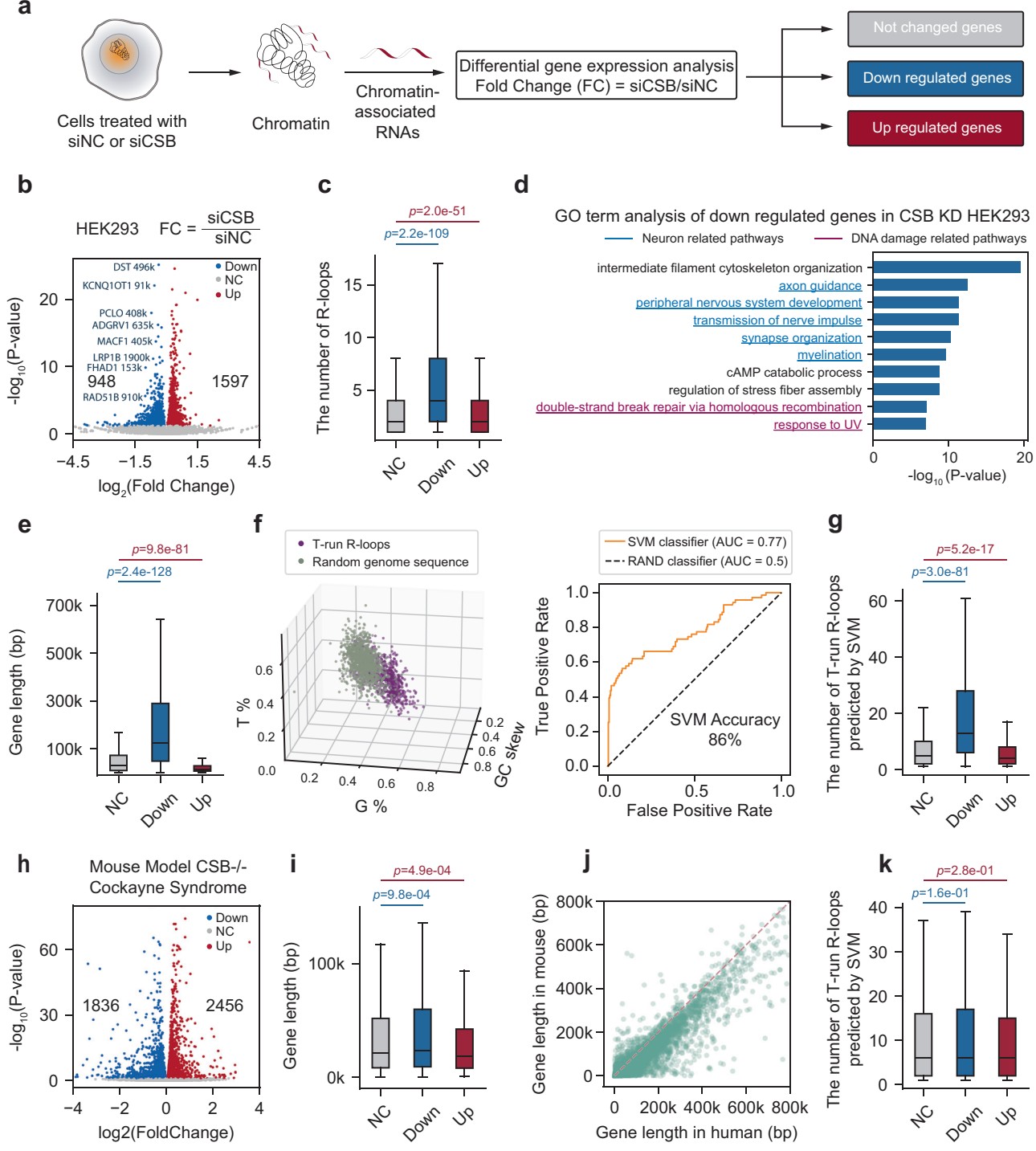

**Fig. 5 | T-run-associated R-loops are enriched in long genes regulated by CSB.** **a** Diagram for chromatin-associated RNA-seq to detect the impact of CSB depletion on transcription. **b** Volcano plot showing differential gene expression based on RNA-seq analysis of chromatin-associated RNAs in siNC and siCSB-treated HEK293 cells. **c** Distribution of the number of R-loops in no change, downregulated and upregulated gene sets. **d** GO term analysis of downregulated genes identified from RNA-seq analysis of chromatin-associated RNAs in siNC and siCSB-treated HEK293 cells. **e** Distribution of gene length in no change, downregulated and upregulated gene sets. **f** 3D scatter plot (left) of three sequence features (G%, T%, and GC skew) with colors representing T-run associated R-loops (purple) and randomly selected sequences across the genome (gray). ROC curves (right) for the SVM classifier on T-run associated R-loops and random genomic sequences. **g** Distribution of the number of predicted T-run-associated R-loops within each gene in no change,

downregulated and upregulated gene sets. **h** Volcano plot of differential gene expression in the kidney of WT and CSB KO mice. **i** Distribution of mouse gene length in no change, downregulated and upregulated mouse gene sets. **j** Scatter plot showing gene length difference between mouse (Y-axis) and human (X-axis) genomes. Each dot represents a conserved gene. **k** Distribution of the number of predicted T-run-associated R-loops within each mouse gene in no change, down-regulated and upregulated mouse gene sets. For the box plot, the center line represents the median, while the upper and lower edges indicate the interquartile range. The whiskers extend to 1.5 times the interquartile range. Statistical sig-nificance was assessed using a two-tailed Mann–Whitney U-test. $n = 25148, 948, 1957$ for unchanged, downregulated, and upregulated genes in plots (**c**, **e**, **g**), and $n = 15415, 1836, 2456$ for unchanged, downregulated, and upregulated genes in plots (**i**, **k**).

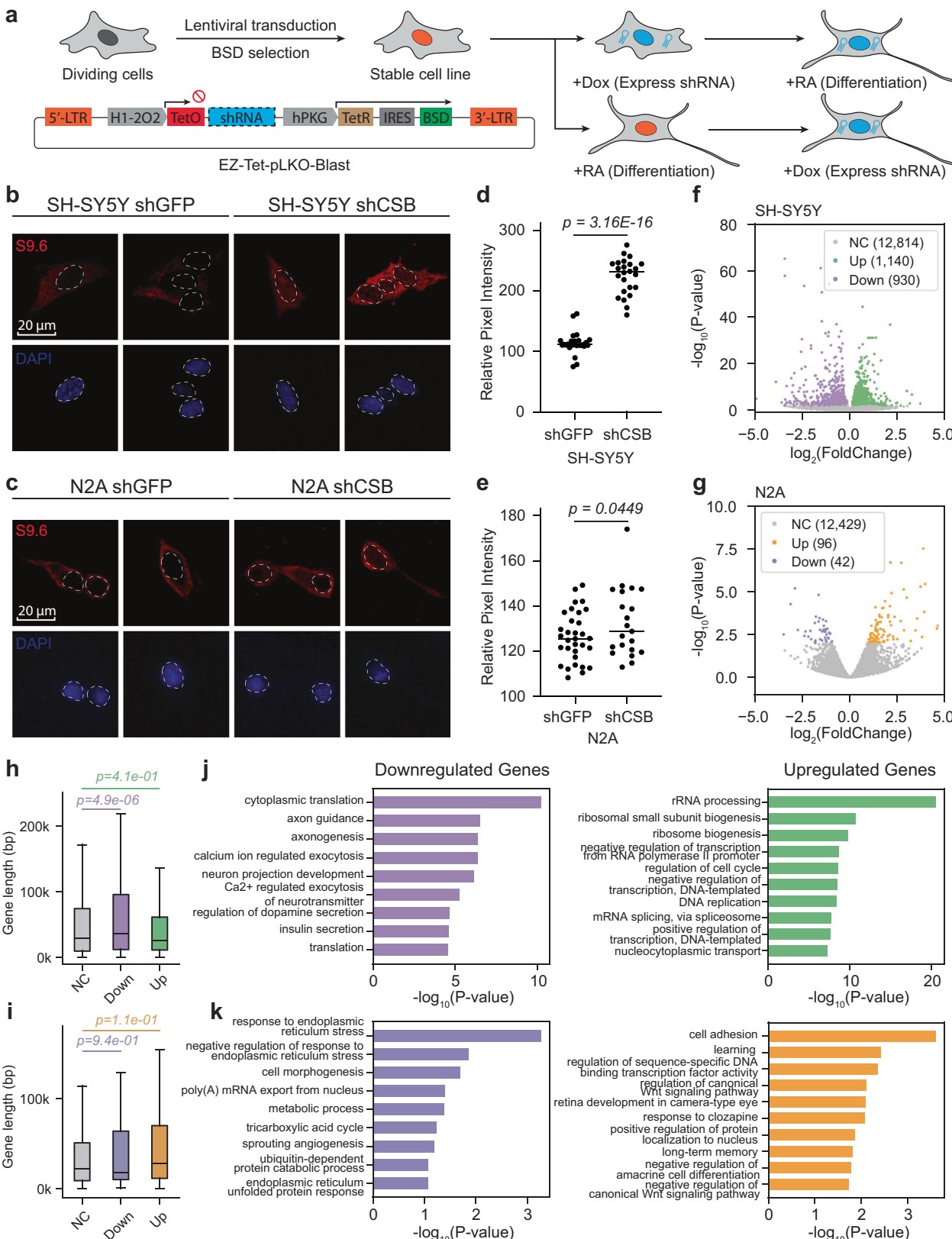

of reduced PRO-seq signals[52]. As shown from our in vitro run-off experiments, the slight reduction of PRO-seq signals likely results from RNAPII backtracking when it transcribes across a T-run. Indeed, multiple early in vitro nuclear run-off studies with purified RNAPII also detected a degree of RNAPII pausing at T-runs, which could be partially mitigated by removing a specific RNAPII subunit Rbp9[53–55].

As depicted in Fig. 7, when coupled with an upstream G-rich sequence, transiently arrested RNAPII at a T-run can trigger R-loop formation because of RNAPII drop-off. This would provide a free RNA end for strand invasion into the DNA bubble formed with the upstream G-rich sequence, as we showed earlier to be necessary and sufficient for de novo R-loop formation within GBs[30]. The formation of R-loops

**Fig. 6 | CSB deficiency selectively impacts on R-loop formation and gene expression on mouse and human neuronal cell lines. a** Overview of the experimental strategy: SH-SY5Y and N2A cells were transduced with lentivirus carrying dox-inducible shGFP and shCSB constructs within the Tet system. Stable cell lines were generated through blasticidin selection. **b, c** Immunofluorescence images displaying the S9.6 (R-loop marker, stained in red) and nuclei (stained in blue) signals in SH-SY5Y (**b**) and N2A (**c**) cells after differentiation by RA for 14 days. Note: Cytosolic S9.6 signals may represent various RNA structures detected by the antibody. **d, e** Quantitative analysis of S9.6 intensity in differentiated SH-SY5Y (**d**) and N2A (**e**) cells. Relative pixel intensity was determined by subtracting the background intensity from the original pixel intensity. Statistical analyses were conducted using a two-tailed unpaired *t*-test (*n* = 30 foci for both shGFP and shCSB treated cells). The *p* values are indicated above the respective graphs. **f, g** Volcano plots illustrating the differential gene expression analysis comparing *CSB* KD versus normal conditions in differentiated SH-SY5Y (**f**) and N2A (**g**) cells. **h, i** Distribution of gene lengths within unchanged, downregulated, and upregulated gene sets in differentiated SH-SY5Y (**i**) and N2A (**j**) cells. For the box plot, the center line represents the median, while the upper and lower edges indicate the interquartile range. The whiskers extend to 1.5 times the interquartile range. Statistical significance was assessed using a two-tailed Mann–Whitney *U*-test, with *n* = 12814, 930, 1140 for unchanged, downregulated, and upregulated genes in plots (**h**), and *n* = 12429, 42, 96 for unchanged, downregulated, and upregulated genes in plots (**i**). **j, k** Gene Ontology (GO) term analysis of downregulated genes (left) and upregulated genes (right) identified from RNA-seq in normal versus *CSB* KD conditions in differentiated SH-SY5Y (**j**) and N2A (**k**) cells.

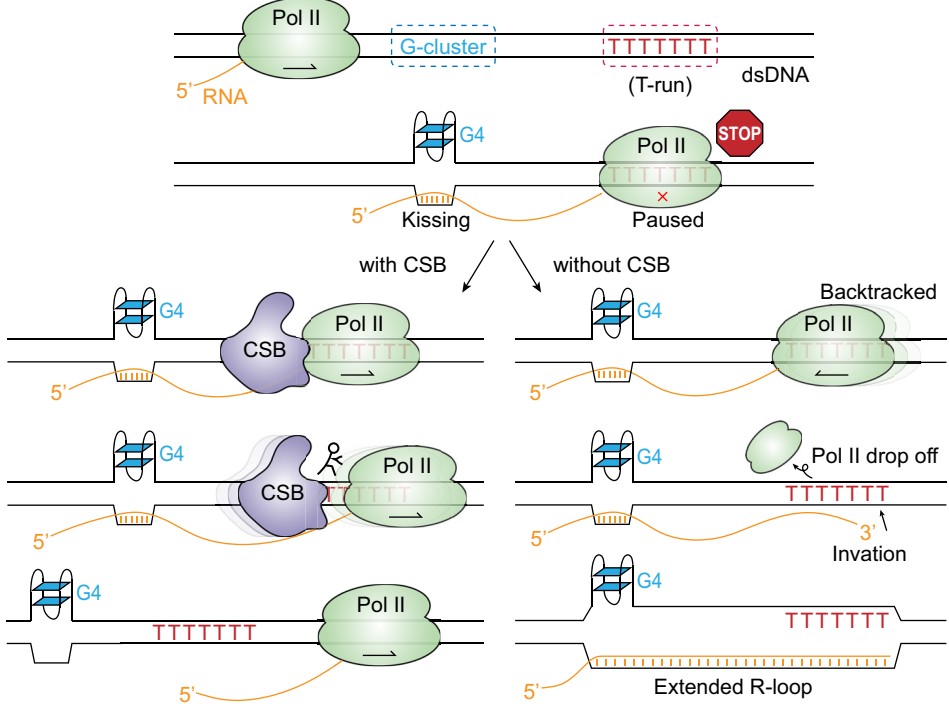

**Fig. 7 | Mechanism of CSB-dependent R-loop formation at T-runs.** CSB is shown in purple, RNAPII in green, G-cluster in blue, and T-run in red. The arrows inside RNAPII indicate the direction of transcription elongation.

would cause additional accumulation of incoming RNAPII. These events appear to frequently occur in the middle of introns rather than at functional 3′ss, which typically lack upstream G-rich sequences.

　　Our results reveal that *CSB* helps prevent transcription backtracking by facilitating RNAPII elongation across the T-run. Indeed, *CSB* might have evolved to establish a mechanism for overcoming such barriers to transcription elongation by pushing RNAPII forward[19], which may be further adapted to deal with other types of barriers induced by DNA-damaging agents, such as UV-induced thymidine dimers and other DNA damage-induced adducts[23]. This would explain the well-characterized function of *CSB* in TC-NER, which may either aid in bypassing minor barriers to restore RNAPII elongation or contribute to the recruitment of DNA repair enzymes as part of the repair process[19].

### A potential mechanism of *CSB* deficiency-associated neurological disorders

Given the broad role of *CSB* in chromatin remodeling, it has been a major puzzle in the field for why *CSB* deficiency imposes selective neuronal vulnerability, which has been implicated in multiple rare autosomal recessive syndromes, such as Cockayne Syndrome, DeSanctis-Cacchione syndrome[56], UV-Sensitive Syndrome[57,58] and Cerebro-Oculo-Facio-Skeletal syndrome 1 (COFS1)[59,60]. While some of these human diseases are clearly related to elevated photosensitivity due to the critical role of *CSB* in TC-NER, other developmental abnormalities could not be solely accounted for by DNA damage-induced events.

　　Our mechanistic studies highlight the crucial role of *CSB* as a regulator of transcription elongation through its direct interaction with RNAPII as an ATP-dependent translocase[19]. Interestingly, as such naturally occurring barriers to transcription elongation are proportionally represented in long genes, the expression of such long genes is selectively impacted by *CSB* deficiency. As most of those long genes are involved in neuronal functions in human cells[44], our findings provide mechanistic insights into how *CSB* deficiency may cause selective neuronal vulnerability in humans. The vulnerability of long genes involved in the nervous system has also been documented in various neurodegenerative diseases, such as ALS linked to the aggregation of the RNA-binding protein TDP-43[61].

### Disease in humans but not in mice because of diverse genome evolution in mammals

Our study reveals many barriers to transcription elongation, especially those in introns, which are effectively mitigated by functional *CSB*. As introns are less conserved than exons during evolution, this explains the varying severity of *CSB* deficiency-induced effects in humans

versus mice. The newly established role of CSB in safeguarding the genome by preventing the unscheduled formation of R-loops aligns well with the cancer-prone phenotype in *CSB* null mice. Interestingly, given that *CSB* deficiency impacts many additional long genes that are selectively expressed and function in human neurons, this discovery points to the molecular basis for understanding severe neurological disorders in individuals with CS. Future studies could focus on examining the expression of neuronal-specific genes in the brain of CS patients, as observed in our cellular models.

In the post-genome era, protein-coding genes are relatively constant in number, but the non-coding part of the genome has been increasingly appreciated for their functions. Genic regions occupy ~20% of the human genome, most corresponding to introns that are removed by splicing to generate mature mRNAs. Besides conserved 5′ ss and 3′ss and other cis-acting splicing regulatory elements, some intronic sequences are known to function as transcription enhancers, but most intronic sequences are thought to have no biological functions. Our current work reveals a key role of a subset of those intronic sequences in transcription control, acting as barriers to RNAPII elongation, and interestingly, their impact on transcription elongation becomes detected only under certain disease conditions.

## Methods

### Plasmid and virus packaging
For HEK293, a lentiviral construct was generated by cloning V5-tagged human RNASEH1 with nuclear localization signal sequence (Addgene plasmid# 111904) into the pFUGW-blast vector (Addgene plasmid# 52962). For N2A and SH-SY5Y, lentiviral constructs were generated by cloning shGFP, shCSB_human, and shCSB_mouse into the EZ-Tet-pLKO vector (Addgene plasmid# 85973).

For virus production, 2.5 μg of pMD2.G, 7.5 μg of psPAX2, and 10 μg of pFUGW-RNaseH1-V5 are transfected into HEK293T cells grown to 50% confluency on a 10 cm dish. After 18 h, media was replaced. The supernatant was harvested at 48- and 72-h post transfection. Viral supernatant is centrifuged at $500 \times g$ to remove cells and debris. It was then concentrated with the Lenti-X concentrator (Clontech Cat# 631231) by the manufacture's instruction. Concentrated viral stock was added to cells in the presence of polybrene (Sigma-Aldrich Cat# TR-1003) at 10 μg/ml. Media was replaced after 24 h. To select the positively infected cells, the cells are added Blasticidin (Thermo Fisher Scientific Cat# R21001) at 10 μg/ml at 48 h from post virus infection. The blasticidin-resistant stable cell line was generated within a week of selection.

Flag-tagged human ERCC6 (CSB) cDNA was cloned into the sleeping beauty transposon system (Addgene plasmid # 60523). To overexpress the CSB flag, HEK293 cells were co-transfected with 9.5 μg of SB CSB construct and 0.5 μg of pSB100X vector. After 24 h, puromycin was added at 1 μg/ml for 2 to 7 days to select positively overexpressed CSB stable cell line.

### Transfection and knockdown
To knockdown CSB, cells were transfected with 20 nM of control siRNA or target siRNA. Cells were harvested at 72 h post transfection. The knockdown efficiency was validated by Western blot. The sequences of the siRNA are listed in supplemental Table S1.

### Neuroblastoma differentiation
Mouse N2A and Human SH-SY5Y cells were seeded onto poly-ᴅ-lysine-coated plates (Gibco Cat# A3890401) and cultured in DMEM + F12 medium (Gibco Cat# 11320033) supplemented with 10% Tet-free FBS and 1% penicillin-streptomycin. Neuronal differentiation of N2A cells was initiated by the addition of Retinoic acid (Sigma-Aldrich Cat# 554720) to the medium, reaching a final concentration of 20 μM[62]. Meanwhile, SH-SY5Y cells underwent neuronal differentiation induced by the addition of Retinoic acid at a final concentration of 10 μM[63]. The

culture medium was refreshed every 3 days to maintain the differentiation process.

### Immunofluorescence and quantification
Cells were cultured on poly-ᴅ-lysine-coated coverslips, subsequently treated, and fixed using 4% paraformaldehyde (PFA) in TBS. Permeabilization was achieved by prechilled methanol treatment, followed by blocking with bovine serum albumin (1 mg/ml) in TBS. Primary antibodies (S9.6, Sigma-Aldrich Cat# MABE1095) were applied, succeeded by secondary antibodies and DAPI staining. Samples underwent mounting and were examined using a CQ-1 microscope (Yokogawa). FIJI software was utilized for processing cell imaging data. Nucleus fluorescence intensity was quantified as the average gray value after subtracting background intensity. Statistical analysis, including mean and *p* value determination, was performed based on data acquired from a minimum of 20 cells per condition.

### Cell viability measurement
Cell survival assessment was conducted using the CellTiter-Glo luminescent viability assay according to the manufacturer's protocol (Promega Cat# G7570). In brief, cells were seeded in 96-well culture plates at a density of 5000 cells per well. Luminescent signals were quantified using a plate reader (Tecan, Spark).

### R-ChIP library preparation
R-ChIP libraries were prepared using our previously described protocol[30,32]. HEK293 cells (10 million cells, grown to 80% confluency) were crosslinked in formaldehyde and then neutralized with glycine. Crosslinked nuclei were isolated, and chromatin was fragmented using a probe sonicator. About 25 μl protein G Dynabeads (Life Technologies Cat# 10009D) and 5 μg V5 antibody (Invitrogen Cat# R96025) conjugate was prepared as per the manufacturer's instruction. To enrich R-loops, magnetic beads and antibody conjugate was added into the fragmented chromatin sample, and then incubated on a cell rotator at 4 °C overnight.

Beads were washed twice with each of the following buffers: TSEI buffer (20 mM Tris-HCl pH 8.0, 150 mM NaCl, 2 mM EDTA, 1% Triton X-100, 0.1% SDS, Protease Inhibitor Cocktail), TSEII buffer B (20 mM Tris-HCl pH 8.0, 500 mM NaCl, 2 mM EDTA, 1% Triton X-100, 0.1% SDS, Protease Inhibitor Cocktail), buffer III (10 mM Tris-HCl pH 8.0, 250 mM LiCl, 1 mM EDTA, 1% NP-40, 1% sodium deoxycholate, Protease Inhibitor Cocktail). It was followed by the wash of TE buffer (10 mM Tris-HCl pH 8.0, 1 mM EDTA) once. Then, chromatin was eluted off the beads in elution buffer (10 mM Tris-HCl pH 8.0, 1 mM EDTA, 1% SDS) at 65 °C for 30 min on a thermomixer. To reverse-crosslink the enriched chromatin, each sample and input chromatin were incubated at 65 °C for 18 h on a thermomixer. Next, the reverse-crosslinked DNA was treated with RNase A (Invitrogen Cat# EN0531) at 37 °C for 2 h, followed by a proteinase K (NEB Cat# P8107S) treatment at 65 °C for 2 h. Phenol:chloroform:isoamyl alcohol extraction was performed twice followed by an ethanol precipitation. The DNA was then resuspended in 20 μl water and used for library construction.

Purified ChIP DNA was constructed into a strand-specific library for Illumina sequencing. First, enriched ssDNA generated from RNase A-digested DNA/RNA hybrid was converted to dsDNA by one cycle extension using Phi29 DNA polymerase (NEB Cat# M0269S) and N9 primer (Table S1). The dsDNA was purified using the DNA cleanup kit (Zymo Cat# D4003) and added a dA tail using Klenow fragment (3′ → 5′ exo-) (NEB Cat# M0212S). DNA was purified again with the DNA cleanup kit and then ligated with the pre-annealed adapters (Table S1). The ligation product was amplified by Phusion polymerase and barcode primers (Table S1) using 14 to 18 cycles depending on the amount of the ChIP DNA. Amplified libraries were size selected using 10% TBE PAGE gel to capture fragments between 150–400 bp. Libraries were quantified using a Qubit kit (Invitrogen Cat# Q32851) and pooled for sequencing.

## ChIP-seq library preparation

ChIP-seq library was prepared as described in protocol[64] with a few modifications. About 10 million HEK293 cells were crosslinked with formaldehyde for 15 min at room temperature. It was followed by 10 min quenching with 125 mM glycine. The crosslinked cells were rinsed twice with 1x PBS twice and then harvested by a cell scraper. Cells were collected by centrifuge at $300 \times g$ for 5 min at 4 °C. The cell pellet was resuspended in 500 µl of sonication buffer (10 mM Tris-HCl pH 8.0, 2 mM EDTA, 0.25% SDS, Protease Inhibitor Cocktail). The chromatin was sonicated by the probe sonicator until most fragments are in the range of 200–700 bp. The cell lysate was diluted with 750 µl of equilibration buffer (10 mM Tris-HCl pH 8.0, 233 mM NaCl, 1 mM EDTA, 1.66% Triton X-100, 0.166% sodium deoxycholate, Protease Inhibitor Cocktail). About 5 µg of antibody and 25 µl magnetic beads conjugate was prepared as described above. Then it was added to the cell lysates and incubated at 4 °C overnight.

Beads were washed twice with each of the following buffers: RIPA-LS buffer (10 mM Tris-HCl pH 8.0, 140 mM NaCl, 1 mM EDTA, 1% Triton X-100, 0.1% SDS, 0.1% sodium deoxycholate, Protease Inhibitor Cocktail), RIPA-HS (10 mM Tris-HCl pH 8.0, 500 mM NaCl, 1 mM EDTA, 1% Triton X-100, 0.1% SDS, 0.1% sodium deoxycholate, Protease Inhibitor Cocktail), RIPA-LiCl (10 mM Tris-HCl pH 8.0, 250 mM LiCl, 1 mM EDTA, 0.5% NP-40, 0.5% sodium deoxycholate, Protease Inhibitor Cocktail). It was followed by the wash of TE buffer (10 mM Tris-HCl pH 8.0, 1 mM EDTA) once.

Then, the beads were resuspended in 25 µl of tagmentation reaction (1x tagmentation buffer, 19 µl H$_2$O, 1 µl Tn5 (Illumina Cat# FC-121-1030)). 5x tagmentation buffer (50 mM Tris-HCl pH 8.0, 25 mM MgCl$_2$, 50% v/v dimethformamide). The reaction was incubated at 37 °C for 10 min in a thermomixer and chilled on ice. The beads were washed twice with RIPA-LS buffer first, followed by the other two washes with 10 mM Tris-HCl pH 8.0. The washed beads were resuspended in 48 µl of ChIP elution buffer (10 mM Tris-HCl pH 8.0, 300 mM NaCl, 5 mM EDTA, 0.4% SDS) and 2 µl proteinase K.

For the input, 19 µl DNA input chromatin was added 1x tagmentation buffer and 1 µl Tn5 enzyme. The tagmentation reaction of input was performed as described above. Then, 48 µl of ChIP elution buffer and 2 µl Proteinase K was added and incubated on ice to inactivate tagmentation reaction.

To reverse-crosslink the enriched and input chromatin, incubate the beads solution and input solution at 55 °C for 1 h and 65 °C for 6–8 h in a thermomixer.

Next, the reverse-crosslinked DNA was purified by ChIP DNA clean and concentration kit (D5205). The DNA was then resuspended in 20 µl water and used for library construction.

## Motif analysis

To identify motifs associated with the head and tail region, motif analysis was performed using findMotifsGenome.pl from homer package with options "-len 12 -p 8 -norevopp -nlen 1 -size 100". Head region and tail region was split based on the summit position of each r-loop. To check the distribution of the enriched motifs' relative summit distance, the corresponding motif position was extracted using findMotifsGenome.pl with options "-find motif_file".

## PRO-seq library preparation

PRO-seq library was prepared as described in the protocol[41] with a few modifications. 10 million HEK293 cells were incubated in swelling buffer (10 mM Tris-HCl pH 8.0, 2 mM MgCl$_2$, 3 mM CaCl$_2$) for 5 min at 4 °C. Cells were collected by centrifuge at $800 \times g$ for 3 min at 4 °C. The cell pellet was resuspended in 5 ml lysis buffer (10% glycerol, 0.5% NP-40, 10 mM Tris-HCl pH 8.0, 2 mM MgCl$_2$, 3 mM CaCl$_2$). Nuclei were collected by centrifuge at $800 \times g$ for 3 min at 4 °C, followed by a wash with 5 ml lysis buffer. After centrifuge, the supernatant was aspirated, and the nuclei pellet was washed in 1 ml freezing buffer (40% glycerol,

5 mM MgCl$_2$, 0.1 mM EDTA, 50 mM Tris-HCl pH 8.0). Then it was centrifuged at $900 \times g$ for 6 min at 4 °C. The nuclei pellet was resuspended in 100 ml freezing buffer. For the global run-on reaction, the nuclei suspension was added 1 volume (100 µl) of reaction mix and incubated at 37 °C for 5 min. The recipe of the reaction mix is listed in supplemental Table S2. After the global run-on reaction, 3 volumes (600 µl) of TRIzol LS (Invitrogen Cat# 10296028) was added and mixed vigorously. Then, RNA was extracted following the manufacturer's instructions. The RNA was resuspended in 20 µl H$_2$O and then denatured at 65 °C on a heat block for 40 s. The denatured RNA was placed on ice and then added 5 µl ice-cold 1 N NaOH and incubated on ice for 10 min. About 25 µl of 1 M Tris-HCl, pH 6.8 was added to quench the fragmentation reaction. Next, buffer exchange was performed by running the fragmented RNA sample through a p-30 column (Bio-Rad Cat# 7326250) according to the manufacturer's instructions.

For each library, 30 µl of streptavidin M280 beads were used. Beads was placed on the magnetic stand for 1 min to remove the supernatant. Beads were washed with 500 µl (0.1 M NaOH, 50 mM NaCl) once, followed by the wash of 500 µl (100 mM NaCl) twice. After removing the supernatant, the beads were resuspended in 150 µl binding buffer (10 mM Tris-HCl pH 7.4, 300 mM NaCl, 0.1% Triton X-100). The RNA sample was added to the prepared 150 beads. The mixture was incubated on a rotator for 30 min at room temperature.

Beads were washed with each of the following buffers: high salt wash buffer once (50 mM Tris-HCl pH 7.4, 2 M NaCl, 0.5% Triton X-100), binding buffer twice (10 mM Tris-HCl pH 7.4, 300 mM NaCl, 0.1% Triton X-100), low salt wash buffer once (5 mM Tris-HCl pH 7.4, 0.1% Triton X-100). The washed beads were resuspended in 300 µl of TRIzol and mixed thoroughly. About 60 µl chloroform was added to the mixture. The aqueous layer was extracted by centrifuge at $14,000 \times g$ for 5 min at 4 °C, and transferred to a new tube. The purified RNA was resuspended in 20 µl H$_2$O and performed the PNK (NEB Cat# M0201S) treatment in a 50 µl reaction. Beads purification was repeated one more time after PNK treatment. The purified RNA was extracted by TRIzol and then constructed into a library using the NEBNext® Multiplex Small RNA Library Prep kit (NEB Cat# E7580S)

## Cell fractionation and RNA-seq library preparation

For the chromatin-associated RNA-seq, chromatin-associated RNA was extracted and used as input of the total RNA-seq library. Chromatin was extracted based on the published protocol (ref) with a few modifications. About 5 million cells were washed twice with ice-cold 1x PBS and then incubated in ice-cold CE buffer (10 mM HEPES-pH 7.5, 0.1 % NP-40, 1 mM EDTA, 1 mM DTT, 60 mM KCl, 340 mM Sucrose) for 5 min on ice. Nuclei were collected by centrifuge at $800 \times g$ for 3 min. The nuclei were rinsed with ice-cold glycerol buffer (20 mM Tris-HCl-pH 7.5, 75 mM NaCl, 0.5 mM EDTA, 0.85 mM DTT, 50% glycerol) and then resuspended in 500 µl glycerol buffer. To lyse the nuclei, the nuclei suspension was mixed with 500 µl nuclei lysis buffer (10 mM HEPES-pH 7.5, 1.0% NP-40, 0.2 mM EDTA, 1 mM DTT, 7.5 mM MgCl$_2$, 300 mM NaCl, and 1 M urea) and incubated on ice for 3 min. The chromatin fraction was collected by centrifuge at $1500 \times g$ for 5 min and then washed twice with ice-cold 1x PBS with 1 mM EDTA. Chromatin-associated RNA was extracted by TRIzol and then constructed into library using NEBNext rRNA Depletion Kit v2 (E7400L) followed by NEBNext Ultra II RNA Library Prep kit (E7775S).

## Sequencing data analysis

Reads of R-ChIP experiments, ChIP-seq, GRO-seq were aligned to the hg38 version of the human reference genome using bwa mem.

For visualization, bigwig files representing read counts across the genome was generated by bamCoverage in deeptools.

To call the peak of each strand in R-ChIP, the bam file was split into plus strand read and minus strand reads using samtools view based on the flag information. Then, for each stranded bam file, because our

strand-specific library was sequenced only on a single end, the macs2 callpeak bimodel cannot be applied flexibly. Therefore, peak was called using the macs2 subcommands based on the instruction (https://github.com/macs3-project/MACS/wiki/Advanced%3A-Call-peaks-using-MACS2-subcommands) For the ChIP-seq, Macs2 callpeak was used with options -p 0.05.

Heatmap and metagene plot was generated using computeMatrix, plotProfile, and plotHeatmap in deeptools.

Reads of RNA-seq were aligned to the hg38 version of the human reference genome using STAR. The read counts of each gene were calculated using featureCounts and hg38 gtf annotation. Differential gene expression analysis was performed using deseq2. For the GO term analysis, the list of upregulated and downregulated genes was input into the DAVID GO analysis web tool (https://david.ncifcrf.gov/tools.jsp) Gene set enrichment analysis was performed using the GSEA linux software followed the instruction (http://www.gsea-msigdb.org/gsea/doc/GSEAUserGuideFrame.html). The SVM used to classify T-run R-loop and random T-run was built using the svm package from sklearn based on the GC-skew, G percentage, and T percentage of each read.

### Protein expression

10-subunits *Saccharomyces cerevisiae* RNAPII was purified essentially as previously described[65]. Briefly, RNAPII with a protein A tag in the N-terminal of the Rpb3 subunit was purified by an IgG affinity column (GE Healthcare), followed by Hi-Trap Heparin (GE Healthcare) and Mono Q anion exchange chromatography columns (GE Healthcare). Recombinant Rpb4/7 heterodimer was purified from Escherichia coli by Ni-affinity chromatography, followed by a gel filtration purification as previously described[66]. Overexpression and purification of Rad26 were performed essentially as described earlier[19].

### Preparation of the T-run sequences containing downstream DNA

The 228 bp DNA fragment containing 178 bp T-run R-loop sequence at the end was amplified from the genomic DNA of the HEK293 cell by two PCR reactions using the primers listed in supplemental Table S1. The 195 bp PCR product was purified by agarose gel electrophoresis. It was then digested by TspRI restriction enzyme and purified by DNA cleanup kit (Zymo Cat# D4003).

### Preparation of T-run sequence containing elongation complex

RNAPII elongation complex (EC10) was prepared essentially as previously described[19]. First, radiolabeled 10-mer RNA was annealed to the template strand DNA(TS), followed by incubation with 10-subunits RNAPII for 10 min at 23 °C and then 2 min at 37 °C. To this, biotin-labeled non-template strand DNA was added and incubated for 5 min at 37 °C, followed by incubation for 20 min at 23 °C. The assembled elongation complex was incubated with streptavidin magnetic beads (NEB Cat# S1420S) for 30 min at 23 °C and subsequently washed five times with elongation buffer (EB) (20 mM Tris-HCl (pH 7.5), 5 mM MgCl$_2$, 40 mM KCl, 5 mM DTT). The immobilized elongation complex was then ligated to the downstream T-run containing sequence by T4 DNA ligase and washed two times with EB buffer. Next, Rpb4/7 was added to the ligation elongation complex at a final concentration of 5 µM and incubated for 10 min at 23 °C to generate a 12-subunit elongation complex, followed by washing three times with EB buffer to remove excess Rpb4/7.

### In vitro transcription assay

Transcription reaction was started by adding 1 mM rNTPs (or specifically indicated in the figure legends) with an additional 3 mM dATP to support Rad26 ATPase activity if Rad26 was added to the assay. The final concentration for Rad26 was 200 nM. Reactions were performed at 30 °C and allowed to continue for the desired time point and then quenched by adding an equal volume of stop buffer (90% formamide, 50 mM EDTA, 0.05% bromophenol blue, and 0.05% xylene cyanol). Samples were denatured for 15 min at 95 °C and the RNA transcripts were separated by 8% denaturing urea PAGE (6 M urea). The gels were visualized by phosphorimaging and quantified using Imagelab software (Bio-Rad). All bands above the ligation truncations (except the transcript from the ligation truncation, labeled with an asterisk) were included in the quantification. The percentage of bypass was calculated as the intensity of run-off bands/total intensity above the ligation truncation.

### In vitro R-loop formation experiments

For R-loop formation experiments, first the immobilized elongation complex was chased by adding rNTPs (concentrations are specifically indicated in the figure and figure legends). After transcription and R-loop formation for 30 min at 30 °C, a final concentration of 0.02 U/µl RNase H (NEB Cat# M0297S) was added, and the reaction was stopped at different time points by adding stop buffer (90% formamide, 50 mM EDTA, 0.05% bromophenol blue and 0.05% xylene cyanol). To increase the specificity of RNase H, a final concentration of 250 mM of NaCl was added to the system before adding RNase H. To evaluate the effect of Rad26 on R-loop formation, a final concentration of 200 nM Rad26 was incubated with the immobilized elongation complex for 10 min before adding rNTPs and dATP to start the reaction. After transcription and R-loop formation for 30 min at 30 °C, a final concentration of 0.02 U/µl RNase H (NEB Cat# M0297S) was added, and the reaction was stopped at different time points by adding stop buffer (90% formamide, 50 mM EDTA, 0.05% bromophenol blue and 0.05% xylene cyanol). Samples were denatured for 15 min at 95 °C and the RNA transcripts were separated by 8% denaturing urea PAGE (6 M urea).

### Quantification and statistical analysis

Statistical parameters were reported in individual figures. The error bar presented in line plots corresponded to mean ± SEM. In the box plot, the median (middle line in the box), first and third quartiles (lower and upper boundaries) and 1.5 times of the interquartile (end of whisker) was plotted. All statistical analyses were performed in Python. Whenever asterisks are used to indicate statistical significance, * stands for $P < 0.05$; ** for $P < 0.01$, and *** for $P < 0.001$.

### Reporting summary

Further information on research design is available in the Nature Portfolio Reporting Summary linked to this article.

## Data availability

The raw data FASTQ files and processed bigwig files have been deposited in or obtained from NCBI GEO database under the accession numbers GSE149760, GSE122736, GSE175792, and GSE226204. Additionally, the brain organoid data was downloaded at: https://www.mdpi.com/article/10.3390/cells13070591/s1.

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

## Acknowledgements

We thank Y. Wang and W. Yuan for technical assistance. J.X. and D.W. were supported by NIH grant (GM102362); X.Z. and Y.H. were supported by NIH grants (HG004659, GM049369, and GM052872); J.H. and H.Q. were supported by National Natural Science Foundation of China (12102086 to J.H.; 82271276 to H.Q.) and by Sichuan Science and Technology Program (2024YFHZ0175 to J.H.; 2022YFS0599 to H.Q.); X.-D.F. was supported by grant (32250710790) from the Natural Science Foundation of China and by funds from Westlake Laboratory, Westlake University, Hangzhou, China.

## Author contributions

X.Z., J.X., D.W., and X.-D.F. designed the experiments. X.Z. performed in vivo cellular and molecular experiments. J.X. conducted in vitro elongation assays. X.Z. performed bioinformatic analysis. J.H. and H.Q. conducted experiments involving neurons differentiated from HDF. X.Z., S.Z., and D.Z. conducted experiments involving N2A and SH-SY5Y cells. Y.H. helped with bioinformatic analysis. X.Z., D.W., and X.-D.F. wrote the paper.

## Competing interests

The authors declare no competing interests.
