## [Peer Review File · Nature Communications]

Cockayne Syndrome linked to elevated R-loops induced by stalled RNA polymerase II during transcription elongationREVIEWER COMMENTS

Reviewer #1 (Remarks to the Author):

In the manuscript entitled "Neurological Cockayne Syndrome Results from R-Loops Induced by Stalled RNA Polymerase II during Transcription Elongation" the authors hypothesized that CSB could be involved in RNAPII elongation. They have performed several experiments providing a possible molecular mechanism for CSB functions.

However, CSB has been already observed to be involved in transcription ref. 19-23, and today it remains unclear how CSB mutations cause neurological disorders.

Furthermore, the authors speculate that the observed mechanisms could be the basis of the differences between human patients and CS mouse models.

The authors did not provide data on human neurons and human progenitors survival/morphology/proliferation/differentiation that could explain the CS disease and a comparison between human and mouse neurons/progenitors is required for the validation of the proposed hypothesis.

Reviewer #2 (Remarks to the Author):

In the manuscript Zhang et al. shows that CSB depletion leads to an upregulation of R-loops in HEK293 cells. They find that a subset (around 10%) R-loops gained in response to CSB depletion has a T-run (enrichment of Ts 3' of the R-loop). They suggest a model where CSB can push RNAPII forward when stalled at these internal T-runs. To validate this model, they perform in vitro transcription assays with the CSB yeast homologue Rad26 (WT and K328R mutant) to show that indeed Rad26 can promote transcription through G clusters followed by a T-run. They suggest that in the absence of CSB, increased RNAPII stalling and stable R-loop formation results in genome instability. The authors further suggest that their observation might explain the neurodevelopmental defect observed in patients with CSB mutations, as neuronal genes are enriched for these T-runs.

The manuscript represents the first report of the TC-NER translocase CSB involvement in resolution of R-loops. In general, the manuscript is well-structured and contains experiments to support most of the authors findings, however in some cases more controls or experiments in additional cell lines are needed. The in vitro transcription assay supports the model of CSB pushing RNAPII through R-loop as proposed based on their initial genome-wide experiments. However, the data on how this might be particularly relevant in neurons and explain the neurodevelopmental and premature ageing defect in CSB patients are still weak and should be expanded upon if they want to support their claims made both in the title and abstract. As it is now there is a disconnect between the results and the title/abstract. Overall, the manuscript provides compelling evidence of a novel role of CSB in resolution of a sub-class of R-loops. This is based on the HEK293 genome-wide data and well as the in vitro data which are both compelling. However, their claims related to the neurological phenotypes needs to be expanded significantly.

Major concerns:

1. The main basis for the paper is that lack of CSB leads to an increase in R-loops genome-wide. However, the only quantification of R-loops upon CSB is done with R-ChIP and it is not clear how this data is normalized. Is this performed with spike-ins to allow for absolute quantification? To really make this point clear and to strengthen the claim that the remaining paper is based on it would be advisable to quantify R-loop by another method in addition to the R-ChIP, such as S9.6 dot blot or image-based quantification.
2. For the in vitro assay a catalytic dead RNaseH1 control should be included to confirm that the RNaseH cleaved bands are indeed that.
3. To make the point that R-loops are a particular problem in neuronal cells they perform chromatin RNA-seq in HEK293 cell. Yes, it is well known that HEK293 shares some features with neuronal cells, but this needs to be confirmed in another cell line if they want to make the strong claim that lack of CSB mediated R-loop resolution is accounting for the neurodevelopmental

defects in CSB patients.

4. The CSB mouse model does not display strong neurodevelopmental defects as human CS patients. The authors want to make the claim that this is due to the lack of T-runs in mouse neuronal genes. However, for this they compare publicly available RNA-seq from CSB $-/-$ mice to their own chromatin RNA-seq from HEK293. Firstly, there is no mention of the tissue type. Secondly, to compare the two they need to use the same RNA-seq approach. This is a good example of how the authors makes very strong claims, but without the proper controls to back them up.

Minor concerns:

1. In connection with the first main point above it would be good to show genome browser tracks of individual R-ChIP replicates (both siNC and siCSB) as in S1G for the merged data.
2. Can the authors use the patient derived CS1AN CSB deficient cell line $-/+$ CSB rescue to verify their speculation that lack of CSB leads to increase levels of R-loops and genome instability?

Reviewer #3 (Remarks to the Author):

Review of Zhang et al. Neurological Cockayne Syndrome Results from R-Loops Induced by Stalled RNA 2 Polymerase II during Transcription Elongation

They describe an intrinsic mechanism by which elongating RNA polymerase II (RNAPII) undergoes transient pausing at internal T-runs where CSB is required to push RNAPII forward. CSB deficiency retards RNAPII elongation in these regions and can lead to genome instability via augmented R-loop formation. They go on to suggest that these processes underlie the severe neurodegeneration in CS.

The paper has not been proofed and there are many semantic errors that should have been fixed. The experiments seem well performed and the Figures are well presented. Unfortunately, the literature background on CS is not up to speed. They mainly cite old literature and not several more recent papers that have appeared on the role of CSB on rDNA and related signaling. Also, many other functions of CSB than its role in NER TCR have been reported and are not mentioned.

Many of the techniques presented have been previously developed and described by this research group and the results that CSB depletion enhances R-Loops in cells are convincing. The observation that these R-loops are formed at T-runs is interesting but mainly confined to specific genes. Using a yeast in vitro transcription system they find a direct role of CSB on pausing and resolution of RNA loops

The work proposing that these above functions lead to neurodegeneration is not convincing. Firstly, they use a kidney cancer cell line to detect changed pathways after CSB kd. While some GO terms related to neuronal functions have changed, Fig 5D, that is not very strong data, and it is surprising that DNA repair and UV sensitivity is not changed more. We know that CSB kd leads to UV sensitivity.

The idea that long genes are more vulnerable to damage is not a new one and a lot of speculation about that has been published. The suggestion that this explains why mice with CS do not have neurodegeneration, because they don't have such large genes, is very simplistic. In general, mice do have different features form humans but some have neurodegeneration.

In all, the demonstration that CSB plays a role in R-loop processing is convincing but not a major new advance, while the suggestion that this underlies neurodegeneration in CS is far from convincing and needs more work. For example, the neurodegeneration in CSB patients is similar to that in CSA patients. If you deplete CSA, are there similar underlying R-loop issues ?

Point-to-Point Response to NCOMMS-23-15112-T

Reviewer #1:

In the manuscript entitled “Neurological Cockayne Syndrome Results from R-Loops Induced by Stalled RNA Polymerase II during Transcription Elongation” the authors hypothesized that CSB could be involved in RNAPII elongation. They have performed several experiments providing a possible molecular mechanism for CSB functions.

However, CSB has been already observed to be involved in transcription ref. 19-23, and today it remains unclear how CSB mutations cause neurological disorders. Furthermore, the authors speculate that the observed mechanisms could be the basis of the differences between human patients and CS mouse models.

The authors did not provide data on human neurons and human progenitors survival/morphology/proliferation/differentiation that could explain the CS disease and a comparison between human and mouse neurons/progenitors is required for the validation of the proposed hypothesis.

We thank the Reviewer for providing the valuable comments. Indeed, the primary aim of our study is centered on understanding genome instability induced by CSB deficiency as the underlying mechanism for CS. Yes, CSB has been implicated in transcription by overcoming certain nucleosome barriers; however, it has been unclear how compromising such a general function would cause distinct functional consequences in mice versus humans.

We acknowledged the importance of a direct comparison between mouse and human neuronal progenitors to validate our hypothesis, as suggested by the Reviewer. We have made substantial efforts in comparing between N2A and SH-SY5Y cells – both widely used cellular models for mouse and human neuronal progenitors.

These cells have the capability to undergo differentiation into neurons upon induction by retinoic acid (RA). We thus asked whether CSB KD might (1) differentially impair RA-induced neuronal differentiation, (2) impact R-loop formation, and (3) alter gene expression on these cells. To this end, we first constructed a lentivirus system to express Tet-inducible shGFP or shCSB to establish Dox-inducible N2A and SH-SY5Y cells. Upon the addition of Dox, we show that we are able to efficiently downregulate CSB in these cells before or after differentiation, as shown in **Fig. R1**.

Fig. R1. Establishment of the doxycycline (dox)-inducible CSB knockdown system on N2A and SH-SY5Y cells.

(a) Overview of the experimental strategy. N2A and SH-SY5Y cells were transduced with a lentivirus carrying Dox-inducible shGFP or shCSB. Stable cell lines were generated through blasticidin (BSD) selection.

(b) Western blot analysis illustrating the impact of dox-induced CSB knockdown on mouse N2A (left) and human SH-SY5Y (right) cells during the proliferative phase or after 14 days of differentiation into non-dividing neuronal-like cells.

Interestingly, we found little impact of CSB KD on both cell types to differentiate into neurons (Fig. R2). Because severe impairment of neuronal differentiation would lead to early lethality in development, these results align well with largely normal development of CSB null mice (van der Horst et al., *Cell* 89, 425-435, 1997) and with the successful differentiation of CS-patient derived ES cells into neurons (Vessoni et al., *Human Mol Genet* 25, 1271-1280, 2016).

Given the fact that CSB null mutations do not dramatically alter the brain structure in both mice and humans, one of the most pressing questions is why CSB deficiency does not compromise the function of mouse brain, but selectively impair the function of human brain to cause severe neurological disorders. Based on our data in the initial submission, we discovered specific sequence features in humans that are responsible for the induction of R-loops, thereby selectively affecting the expression of long genes associated with neuronal functions, but such features are largely missing in mouse cells. Our study is the first report of such distinction between mice and humans in the vast CSB literature.

Fig. R2. Morphological and growth analysis during differentiation of CSB KD N2A and SH-SY5Y cells.

(a) Microscopic images depicting morphological changes during retinoic acid (RA)-induced differentiation of N2A and SH-SY5Y cells into neuronal-like cells, comparing cells treated with shGFP (control) versus shCSB (CSB KD).

(b) Quantification of live N2A (left) and SH-SY5Y (right) cells throughout the differentiation process to show that these have switched from the dividing to the non-dividing state.

In response to the reviewer's request for further validation of the observed difference between mouse and human cells, we compared and quantified R-loop formation between RA-differentiated N2A and SH-SY5Y cells. As expected, Dox-induced CSB KD significantly triggered R-loop formation in differentiated SH-SY5Y cells, but not N2A cells, as evident from the staining with S9.6, a monoclonal antibody specifically targeting RNA-DNA hybrids within the nucleus (Fig. R3).

Fig. R3. Quantification of R-Loops in the nucleus by S9.6 Staining following CSB knockdown in differentiated N2A and SH-SY5Y cells.

(a) Schematic representation of the experimental design to evaluate R-loop levels in neurons derived from differentiated N2A and SH-SY5Y cells.

(b and d) Immunofluorescence images displaying S9.6 (R-loop marker, stained in red) and nuclei (stained in blue) signals in differentiated N2A (b) and SH-SY5Y (d) cells. Note: Cytosolic S9.6 signals may represent various RNA structures cross reacted with the antibody, and thus, not considered.

(c and e) Quantitative analysis of S9.6 staining intensity in differentiated N2A (c) and SH-SY5Y (e) cells. Statistical analyses were conducted using a two-tailed unpaired t-test ($n = 30$). The p-values are indicated above the respective graphs.

Concurrently, we also performed RNA-seq on these differentiated neuronal progenitors (Fig. R4). Consistent with our observations in HEK293 cells, we found that CSB KD in SH-SY5Y cells preferentially down-regulated relatively long genes, a pattern notably absent in N2A cells. Furthermore, the impact of CSB KD on N2A cells was markedly less significant (~ 8 -fold lower $\log_{10}P$ -values) in gene length distributions and less substantial in terms of gene expression changes (>10 -fold fewer affected genes). Most importantly,

we found that downregulated genes were linked to neuronal functions in CSB-knockdown SH-SY5Y cells, not in N2A cells, based on our Gene Ontology (GO) term analysis, which upregulated genes in both N2A and SH-SY5Y cells did not exhibit any enrichment in neuronal function pathways.

Non-dividing N2A

Non-dividing SHSY5Y

Fig. R4. RNA-Seq analysis of CSB knockdown effects in differentiated mouse N2A and human SH-SY5Y Cells.

(a and e) Volcano plots illustrating differential gene expression under CSB knockdown versus normal conditions in differentiated N2A (a) and SH-SY5Y (e) cells.

(b and d) Gene Ontology (GO) term analysis of down-regulated genes identified from RNA-seq under normal versus CSB knockdown conditions in differentiated N2A (b) and SH-SY5Y (d) cells.

(c and f) Distribution of gene lengths within unchanged, downregulated, and upregulated gene sets identified in differentiated N2A (c) and SH-SY5Y (f) cells.

(d and h) GO term analysis of up-regulated genes identified from RNA-seq under normal versus CSB knockdown conditions in differentiated N2A (d) and SH-SY5Y (h) cells.

Together, these findings reinforce the selective impact of CSB deficiency on human neuron progenitors. We have included these data in new Fig. 6 (R3 and R4) and Extended Data Fig.S12 (R1 and R2) in the revised manuscript.

Reviewer #2:

In the manuscript Zhang et al. shows that CSB depletion leads to an upregulation of R-loops in HEK293T cells. They find that a subset (around 10%) R-loops gained in response to CSB depletion has a T-run (enrichment of Ts 3' of the R-loop). They suggest a model where CSB can push RNAPII forward when stalled at these internal T-runs. To validate this model, they perform in vitro transcription assays with the CSB yeast homologue Rad26 (WT and K328R mutant) to show that indeed Rad26 can promote transcription through G clusters followed by a T-run. They suggest that in the absence of CSB, increased RNAPII stalling and stable R-loop formation results in genome instability. The authors further suggest that their observation might explain the neurodevelopmental defect observed in patients with CSB mutations, as neuronal genes are enriched for these T-runs.

The manuscript represents the first report of the TC-NER translocase CSB involvement in resolution of R-loops. In general, the manuscript is well-structured and contains experiments to support most of the authors findings, however in some cases more controls or experiments in additional cell lines are needed. The in vitro transcription assay supports the model of CSB pushing RNAPII through R-loop as proposed based on their initial genome-wide experiments. However, the data on how this might be particularly relevant in neurons and explain the neurodevelopmental and premature ageing defect in CSB patients are still weak and should be expanded upon if they want to support their claims made both in the title and abstract. As it is now there is a disconnect between the results and the title/abstract. Overall, the manuscript provides compelling evidence of a novel role of CSB in resolution of a sub-class of R-loops. This is based on the HEK293T genome-wide data and well as the in vitro data which are both compelling. However, their claims related to the neurological phenotypes needs to be expanded significantly.

We thank the Reviewer for thoughtful comments and valuable feedback on our manuscript. We are delighted that the Reviewer concisely summarized our findings, highlighting "*The manuscript represents the first report of the TC-NER translocase CSB*

involvement in resolution of R-loops. In general, the manuscript is well-structured and contains experiments to support most of the authors findings."

Additionally, the Reviewer pointed out specific claims in our title and abstract that need further experimental support. We greatly appreciated these constructive critiques and made concerted efforts to turn down our title and abstract to some extent. More importantly, we have experimentally addressed individual concerns raised by including additional controls and extending our findings to additional cellular models.

Major concerns:

1. The main basis for the paper is that lack of CSB leads to an increase in R-loops genome-wide. However, the only quantification of R-loops upon CSB is done with R-ChIP and it is not clear how this data is normalized. Is this performed with spike-ins to allow for absolute quantification? To really make this point clear and to strengthen the claim that the remaining paper is based on it would be advisable to quantify R-loop by another method in addition to the R-ChIP, such as S9.6 dot blot or image-based quantification.

Regarding the quantification of R-ChIP signals, we did not use the spike-in approach for determining absolute signal intensities on individual peaks. Instead, we processed our data following the well-established pipeline in most published ChIP-seq experiments by using Counts Per Million (CPM) values for genome-wide normalization of ChIP-seq datasets, enabling the identification of R-ChIP peaks and facilitating comparison between different experimental conditions. Of note, like ChIP-seq, R-ChIP targets specific genomic hotspots where R-loops form, occupying only a small fraction of total sequence reads, while the majority of reads constitutes the background.

More importantly, in response to the suggestion of using an independent approach to validate elevated R-loops, we performed S9.6 staining as suggested on HEK293T cells. This complementary image-based quantification conducted alongside R-ChIP analysis demonstrates increased S9.6 signals subsequent to CSB KD (**Fig. R5**), providing additional support to our conclusions, as suggested. These data have been included as Extended Data **Fig. S2** in the revised manuscript.

Fig. R5. Detection of elevated R-loops in HEK293T cells in response to CSB KD based on S9.6 staining signals in the nucleus.

(a) S9.6 staining in mock-treated and siCSB treated cells.

(b) Quantification of S9.6 signals in nuclei of mock-treated and siCSB treated cells.

2. For the in vitro assay a catalytic dead RNaseH1 control should be included to confirm that the RNaseH cleaved bands are indeed that.

We decided not conducting the suggested experiment with a catalytic dead RNaseH1 for two reasons. First, implementing the suggested experiment would demand a complete repetition of the entire set of experiments and run in parallel with both active and catalytic dead RNaseH1 in order to make a meaningful comparison. Second, we suppose that the suggested experiment aims to address a potential problem that the cleaved bands might result from the action of certain contaminated ribonucleases in our RNaseH1 prep. We believe that our existing data have already offset such possibility because none of RNA bands in our control sample was affected by the treatment of active RNaseH1. Furthermore, not all bands in our RNase H1-treated samples were degraded into smears, indicating the enzyme is acting specifically on RNA:DNA hybrids in regions enriched with typical G and T-run. Therefore, we've chosen to prioritize our focus on executing other critical experiments as recommended.

3. To make the point that R-loops are a particular problem in neuronal cells they perform chromatin RNA-seq in HEK293T cell. Yes, it is well known that HEK293T shares some features with neuronal cells, but this needs to be confirmed in another cell line if they want to make the strong claim that lack of CSB mediated R-loop resolution is accounting for the neurodevelopmental defects in CSB patients.

We acknowledge the importance of this suggestion and have therefore extended our analysis to include commonly used human neuronal progenitor SH-SY5Y cells with the mouse neuronal progenitor N2A cells as control. As shown above in addressing a set of

related questions raised by Reviewer #1, we provide the following additional data (the figures are not duplicated to avoid redundancy):

- (1) Dox-induced depletion of CSB in SH-SY5Y and N2A cells before or after RA-induced neuronal differentiation (Fig. R1),
- (2) Lack of detectable impact of CSB KD on neuronal differentiation on this pair of cellular models (Fig. R2),
- (3) Selective induction of R-loops in RA-differentiated human SH-SY5Y cells, but not mouse N2A cells (Fig. R3).
- (4) Differential impact on long genes with neuronal functions in SH-SY5Y cells, but not mouse N2A cells (Fig. R4).

Together, we believe that these additional experiments have substantially strengthened our conclusions. These data have been incorporated into new Fig. 6 and Extended Data Fig. S12 in the revised manuscript.

4. The CSB mouse model does not display strong neurodevelopmental defects as human CS patients. The authors want to make the claim that this is due to the lack of T-runs in mouse neuronal genes. However, for this they compare publicly available RNA-seq from CSB $-/-$ mice to their own chromatin RNA-seq from HEK293T. Firstly, there is no mention of the tissue type. Secondly, to compare the two they need to use the same RNA-seq approach. This is a good example of how the authors makes very strong claims, but without the proper controls to back them up.

While stating the source of CSB knockout mice, we inadvertently omitted mentioning the tissue type used to generate the RNA-seq data, which in this case is kidney cells from CSB $-/-$ mice.

Importantly, in the revised manuscript, we've made efforts to ensure a more reliable comparison by employing the same RNA-seq approach when analyzing SH-SY5Y and N2A cells, which has yielded consistent conclusions, as depicted in Fig. R4.

Minor concerns:

1. In connection with the first main point above it would be good to show genome browser tracks of individual R-ChIP replicates (both siNC and siCSB) as in S1G for the merged data.

We showed the merged data because the tracks of individual replicates exhibit similarity. As suggested, we chose to display the merged tracks in Fig. 1b and tracks of replicates in fig. S1G (instead of a different gene example as shown in original fig. S1G).

2. Can the authors use the patient derived CS1AN CSB deficient cell line $-/+$ CSB rescue to verify their speculation that lack of CSB leads to increase levels of R-loops and genome instability?

We have not conducted the suggested experiment on CS1AN cells because the cells are fibroblasts from a CS patient, which do not efficiently express many of those neuronal-

specific long genes. Considering potential limitations in robust R-loop profiles for comparison in these fibroblast lines, we redirected our efforts towards expanding our analysis. Specifically, we have focused on human neuronal progenitor SH-SY5Y cells and their mouse counterpart N2A cells. We hope the Reviewer agrees with this rationale.

Additionally, we also extended our analysis to human dermal fibroblasts (HDFs) where the RNA binding protein PTBP1 was knocked down to induce neurogenesis in these cells, as we previously reported (Xue et al., *Cell* 152:82-86, 2013). These HDFs closely mirror CS1AN cells, as both originating from human skin fibroblasts. These new data on neurons differentiated from HDFs are well aligned with our observations on HEK293 cells, including CSB KD-induced R-loop formation detected with S9.6 staining, differential impact on long genes as determined by both metagene analysis and with a specific example on RAD51b as shown in Fig. R6. These data have been included as Extended Data Fig.S10 in the revised manuscript.

Fig. R6. Impact of CSB knockdown on long genes in PTBP1 depleted Human Dermal Fibroblasts (HDFs).

(a) Immunofluorescence images displaying S9.6 signals (R-loop marker, stained in red) and nuclei (stained in blue) in neurons derived from HDFs. Note: Cytosolic S9.6 signals may represent various RNA structures cross-reacted with the antibody.
 (b) Quantitative analysis of S9.6 staining intensity in PTBP1 KD HDFs. Statistical analyses were conducted using a two-tailed unpaired t-test (n = 20). The p-values are indicated above the respective graphs.
 (c) Volcano plot displaying the differential gene expression analysis before and after CSB knockdown in HDFs.

- (d) Gene length distribution within unchanged, downregulated, and upregulated gene sets.
- (e) Genome browser tracks demonstrating changes in RNA-seq signals on a relative long gene RAD51B in comparison with a short multiple-exon gene ZFYVE26 in HDFs treated with shGFP versus shCSB. Gene annotations are indicated above the tracks.

Reviewer #3:

Review of Zhang et al. Neurological Cockayne Syndrome Results from R-Loops Induced by Stalled RNA 2 Polymerase II during Transcription Elongation

They describe an intrinsic mechanism by which elongating RNA polymerase II (RNAPII) undergoes transient pausing at internal T-runs where CSB is required to push RNAPII forward. CSB deficiency retards RNAPII elongation in these regions and can lead to genome instability via augmented R-loop formation. They go on to suggest that these processes underlie the severe neurodegeneration in CS.

The paper has not been proofed and there are many semantic errors that should have been fixed. The experiments seem well performed and the Figures are well presented.

We thank the Reviewer for providing the valuable comments. We appreciate the Reviewer's positive assessment of the experiments and the presentation of our figures. We sincerely apologize for the errors in describing our observations. We have carefully proofread and revised the manuscript to ensure its accuracy.

Unfortunately, the literature background on CS is not up to speed. They mainly cite old literature and not several more recent papers that have appeared on the role of CSB on rDNA and related signaling. Also, many other functions of CSB than its role in NER TCR have been reported and are not mentioned.

We originally cited CSB-related literature that is relevant to our current studies. The Reviewer suggested us to include more recent papers on CSB functions, such as rDNA and related signaling, besides NER-TCR. Accordingly, we have included additional references to provide an overview on CSB's diverse functions, with particular emphasis on its role in rDNA expression through binding to nucleolin to enhance rRNA synthesis. This appears related to the most enriched GO term on cytoplasmic translation among downregulated genes in CSB KD SH-SY5Y cells (see Fig. R4). Intriguingly, our observations also reveal rRNA processing as the most enriched GO term among up-regulated genes, likely reflecting a compensatory response to the effects induced by CSB deletion (see Fig. R4)

Many of the techniques presented have been previously developed and described by this research group and the results that CSB depletion enhances R-Loops in cells are convincing. The observation that these R-loops are formed at T-runs is interesting but mainly confined to specific genes. Using a yeast in vitro transcription system they find a direct role of CSB on pausing and resolution of RNA loops

The work proposing that these above functions lead to neurodegeneration is not convincing. Firstly, they use a kidney cancer cell line to detect changed pathways after CSB KD. While some GO terms related to neuronal functions have changed, Fig 5D, that is not very strong data, and it is surprising that DNA repair and UV sensitivity is not changed more. We know that CSB KD leads to UV sensitivity.

We are pleased that the Reviewer is convinced by our data on induced R-loops in CSB KD cells. However, the Reviewer expressed concerns regarding the relevance of the data derived from HEK293 cells. Additionally, the Reviewer wondered why genes associated with DNA repair and UV sensitivity were not enriched in our GO terms analysis.

To rigorously address the concerns on drawing some of our conclusions solely from the data from HEK293 cells, we have extended our analysis to human SH-SY5Y cells in comparison with mouse N2A cells — both widely used as cellular models for neuronal progenitors. As expected, we again observed a selective impact of CSB KD on relatively long genes associated with neuronal functions in differentiated human SH-SY5Y cells, but not in differentiated mouse N2A cells, as presented in Fig. R4. To complement the observation on CSB-rescued CS1AN human fibroblasts, we also extended our analysis to PTBP1-depleted human dermal fibroblasts (HDFs), showing both induced S9.6 signals and the compromised expression of relatively long genes as shown in Fig. R6. We have included these data in new Fig. 6 and Extended Data Fig. S10 and Fig. S12 in the revised manuscript.

Regarding enriched genes related to DNA repair and UV sensitivity, we wish to point out that we did detect terms associated with double-strand repair and UV response, but these terms were ranked lower in our analysis (9th and 10th place in original Fig. 5d), likely because we did not subject our cells to UV treatment.

The idea that long genes are more vulnerable to damage is not a new one and a lot of speculation about that has been published. The suggestion that this explains why mice with CS do not have neurodegeneration, because they don't have such large genes, is very simplistic. In general, mice do have different features from humans but some have neurodegeneration.

We agree that attributing severe neurological disorders in humans solely to the impact on longer genes may oversimplify the situation. There are likely other contributing features that account for the differences observed between mice and humans. In the revised manuscript, we have adjusted several statements to emphasize that the selective impact on long genes may be one of the contributing factors to the diverse phenotypes observed between CSB-deficient mice and humans.

In all, the demonstration that CSB plays a role in R-loop processing is convincing but not a major new advance, while the suggestion that this underlies neurodegeneration in CS is far from convincing and needs more work. For example, the neurodegeneration in CSB patients is similar to that in CSA patients. If you deplete CSA, are there similar underlying R-loop issues?

We do believe that CSB deficiency-induced R-loops represent a key advance in the field, as also pointed out by Reviewer #2. However, we acknowledge the Reviewer's valid concern about the limitations of proposing a neurodegeneration mechanism in CS based primarily on data from HEK293 cells. In this revised manuscript, we have made substantial efforts in addressing this critique by conducting additional experiments on differentiated human SH-SY5Y cells in comparison with mouse N2A cells. The new data significantly reinforce the conclusions drawn from our initial findings on HEK293 cells. We believe that the Reviewer will be pleased to see a series of new evidence presented in the revised manuscript.

Regarding CSA and its potential role in R-loop formation and resolution compared to CSB, we recognize it as an intriguing and important avenue for further exploration. However, in this current revision, we have not allocated resources and manpower to pursue this new direction simply because we believe it deserves a comprehensive investigation as part of future studies. We have therefore focused on addressing the more immediate and critical questions raised by all three reviewers.

REVIEWER COMMENTS

Reviewer #1 (Remarks to the Author):

The authors have performed several experiments on mouse and human cell lines. However, the proposed data are not sufficient to validate their hypothesis.

The sentence "For this purpose, we utilized a pair of commonly used mouse and human neuronal progenitors, N2A (mouse) and SH-SY5Y (human) cells" is not correct. The N2A and SH-SY5Y cells are mouse and human cell lines, therefore could contain several mutations and genomic alterations that could explain the observed phenotypes (also after differentiation). The authors should use a primary culture of mouse neurons and more reliable human neurons (like iPSc-derived neurons) or at least not cancer cell lines.

Reviewer #2 (Remarks to the Author):

The authors have adequately addressed my concerns.

Point-to-Point Response to NCOMMS-23-15112B

Reviewer #1:

The authors have performed several experiments on mouse and human cell lines. However, the proposed data are not sufficient to validate their hypothesis.

The sentence "For this purpose, we utilized a pair of commonly used mouse and human neuronal progenitors, N2A (mouse) and SH-SY5Y (human) cells" is not correct. The N2A and SH-SY5Y cells are mouse and human cell lines, therefore could contain several mutations and genomic alterations that could explain the observed phenotypes (also after differentiation). The authors should use a primary culture of mouse neurons and more reliable human neurons (like iPSc-derived neurons) or at least not cancer cell lines.

We have replaced "neuronal progenitors" with "cell lines that have the properties of neuronal progenitors" to describe N2A and SH-SY5Y cells.

We would like to stress that we not only performed analysis on several mouse and human cell lines, but also on neurons derived from normal cells, such as human dermal fibroblasts (HDF). Despite inherent variabilities among different cell lines, we consistently observed the same effect across all tested lines.

The reviewer recommended additional analysis of "use primary culture of mouse neurons and more reliable human neurons (like iPSC-derived neurons)". As total CSB knockout showed little impact on brain development, which entirely agree with our observations on cell lines, we do not believe that it is necessary to further elaborate on primary cells. With respect to extending our analysis to "more reliable human neurons, such as iPSC-derived neurons, we are pleased to note a newly deposited manuscript in bioRxiv from the James Adjaye lab that has exactly provided the data for our purpose.

In this study, the authors analyzed two Cockayne Syndrome (CS) patient-derived iPSCs (CS789 and IUFi100) in comparison with a health control (B4) iPSC line. They first differentiated these cells into neuronal progenitors (neurospheres) and then cerebral organoids. As expected, CSB deficiency showed little to no effect on neuronal differentiation, mirroring our observations in both human SH-SY5Y and mouse N2A cells. Through RNA-seq, they identified a large cohort of up- and down-regulated genes in response to CSB mutations. Through KEGG and GO term analysis, they found that while up-regulated genes are related to general cellular functions, down-regulated genes are predominantly involved in various neuronal functions. This parallels what we observed in neurons differentiated from human SH-SY5Y cells or trans-differentiated from human dermal fibroblasts.

Although these authors described the CSB-deficiency-induced phenotype, they do not know the mechanism, which is elucidated in our current work. We have thus downloaded their data from the public domain and plotted the length of affected genes.

As shown in extended Figure 10, panel f, we observed that the down-regulated genes in both CSB-deficient neurons relative to the healthy control are much longer compared to the upregulated ones, aligning perfectly with our findings. We have now incorporated this latest information in our revised manuscript, as yellow-highlighted in the main text and relevant figure legends.

Reviewer #2:

The authors have adequately addressed my concerns.

We would like to thank Reviewer #2 for taking the time to review our manuscript and for acknowledging that we have adequately addressed their concerns.

Reviewer #3:

This reviewer provided a confidential comment to editor on a suggestion similar to what Reviewer #1 specifically suggested. We have thus completely addressed the request, as detailed above.

REVIEWERS' COMMENTS

Reviewer #1 (Remarks to the Author):

The authors did not address my questions.

They did not provide convincing data to validate their hypothesis in neurons.

I would remove all the sentences with putative neuron effects, explanation, and discussion.

Point-to-Point Response to NCOMMS-23-15112B

Reviewer #1:

The authors did not address my questions.
They did not provide convincing data to validate their hypothesis in neurons.
I would remove all the sentences with putative neuron effects, explanation, and discussion.

We thank the Reviewer for taking the time to review our manuscript. We would like to emphasize that our current study has been focused on understanding the molecular basis for CSB to protect genome stability. Our studies reveal selective impact of CSB deficiency on relatively long genes that are selectively expression and function in neurons of human cells, but not mouse cells, thus providing critical insights into different phenotypes caused by CSB null mutation in humans vs mice.

The Reviewer has been insisting us to validate our hypothesis on iPSC-derive human neurons. In our last revision, we provided our analysis of the independent RNA-seq data recently deposited in BioRxiv on iPSC-derived human neurons, which further validated our conclusions. We thus believe that we have fully addressed the question of this reviewer.

With respect to removing “all sentences on putative neuron effects, explanation, and discussion”, as suggested by the Reviewer and editor, we have carefully gone through the entire manuscript to avoid any potential overstatements, particularly removing “claims of impact on mouse and human neuronal progenitors”.